# Precise atom-to-atom mapping for organic reactions via human-in-the-loop machine learning

Shuan Chen [1,2], Sunggi An[1,2], Ramil Babazade [3] & Yousung Jung [1,2,4,5] ✉

Atom-to-atom mapping (AAM) is a task of identifying the position of each atom in the molecules before and after a chemical reaction, which is important for understanding the reaction mechanism. As more machine learning (ML) models were developed for retrosynthesis and reaction outcome prediction recently, the quality of these models is highly dependent on the quality of the AAM in reaction datasets. Although there are algorithms using graph theory or unsupervised learning to label the AAM for reaction datasets, existing methods map the atoms based on substructure alignments instead of chemistry knowledge. Here, we present LocalMapper, an ML model that learns correct AAM from chemist-labeled reactions via human-in-the-loop machine learning. We show that LocalMapper can predict the AAM for 50 K reactions with 98.5% calibrated accuracy by learning from only 2% of the human-labeled reactions from the entire dataset. More importantly, the confident predictions given by LocalMapper, which cover 97% of 50 K reactions, show 100% accuracy for 3,000 randomly sampled reactions. In an out-of-distribution experiment, LocalMapper shows favorable performance over other existing methods. We expect LocalMapper can be used to generate more precise reaction AAM and improve the quality of future ML-based reaction prediction models.

Atom-to-atom mapping (AAM) plays a crucial role in preparing reaction data by identifying the one-to-one mapping between reactant atoms and product atoms. High-quality AAM allows fast recognition of the reaction center of a given chemical reaction, which is essential for many of the developed methods working on chemical reaction analysis and prediction.

One of the widely used applications of AAM is the construction of a condensed graph of reaction (CGR)[1,2], which combines the reactant and product graphs into a single representation and has shown promise in various reaction tasks, including reaction condition prediction[3,4], reaction similarity search[5], and even predicting advanced reaction quantities such as activation energy or reaction yield[6,7]. Additionally, AAM enables the automatic identification of reaction centers and extraction of reaction templates from

databases, which are utilized in predicting reaction outcomes[8–10] and single-step retrosynthesis[11–15] machine learning (ML) models. Since these applications are highly dependent on the AAM of reaction data, the quality of AAM greatly impacts the performance of machine learning models. For instance, the incorrect mapping on an alkene epoxidation would generate an invalid retrosynthesis reaction template and unclear reaction mechanism (Fig. 1), turning valuable reaction data into noisy reaction data. Unfortunately, the commonly used USPTO reaction dataset has been reported to contain issues like incorrect AAM or missing reactants, which directly affect downstream ML models[10,16–18]. Incorrect AAM can lead to the learning of incorrect chemistry, resulting in unrealistic prediction models and retrosynthesis pathways. With the growing number of downstream models being developed, the curation of high-quality AAM for

[1]Department of Chemical and Biomolecular Engineering, KAIST, Daejeon, South Korea. [2]Department of Chemical and Biological Engineering, Seoul National University, Seoul, South Korea. [3]Graduate School of AI, KAIST, Daejeon, South Korea. [4]Institute of Chemical Processes, Seoul National University, Seoul, South Korea. [5]Interdisciplinary Program in Artificial Intelligence, Seoul National University, Seoul, South Korea. ✉e-mail: yousung.jung@snu.ac.kr

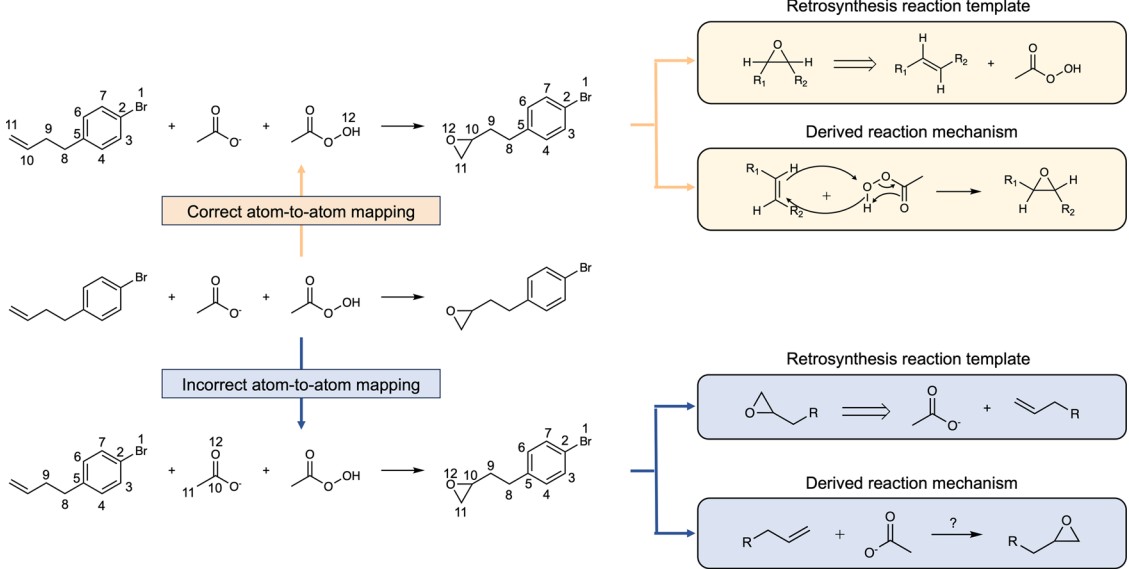

**Fig. 1 | The importance of accessing correct atom-to-atom-mapping (AAM) in terms of generating retrosynthesis reaction templates and deriving reaction mechanisms based on chemical knowledge.** The upper AAM is generated by the proposed AAM model LocalMapper, and the bottom AAM is recorded in the USPTO reaction dataset.

reaction datasets becomes an urgent task to ensure the quality of reaction prediction models.

Existing methods for AAM identification can be generally categorized into rule-based[17,19–25] and ML-based methods[18,26]. Most of the rule-based methods identify AAM based on minimal chemical distance (MCD)[27] or maximum common subgraph (MCS) isomorphism algorithms[28]. Solving the AAM problem by rule-based approaches is challenging because such a subgraph isomorphism problem has been known be be an NP-hard problem since the 1970s, and there is no efficient algorithm to find the exact solutions[29–31]. Recently, machine learning models have been developed to bypass the time-consuming subgraph matching process and map the atoms in the reactions directly by the information extracted from the model's learned features. Schwaller et al.[18] proposed an unsupervised-learning-based model called RXNMapper to link the grammar dependency of each atom between reactants and products. By focusing on specific attention weights of the language model, RXNMapper not only achieved a promising prediction accuracy that outperformed existing rule-based methods but also largely reduced the computational time of performing AAM on large reaction datasets. More recently, Nugmanov et al.[26] developed GraphomerMapper using a similar unsupervised learning strategy with RXNMapper based on a graph-based Transformer and trained the model on a much larger reaction dataset.

Although the above-mentioned approaches have shown improving accuracy over previous methods, a perfect 100% accuracy of AAM is required since the flaw in the reaction data will be amplified in the downstream reaction prediction models. Yet, currently, existing methods have not shown a reliable approach to detect potentially incorrectly predicted AAM, which makes the error in the predictions hard to identify. Furthermore, although existing ML-based unsupervised methods are found to be much faster than rule-based methods and applicable to a wider range of reactions, training a model without knowing the correct AAM may lead to unexpected errors even for simple reactions. As later shown in this paper, previous methods have incorrectly mapped over 5% of reactions in the widely used US patent dataset.

Here, we present a precise graph-based AAM model, named *LocalMapper*, via human-in-the-loop machine learning. Apart from previous ML-based approaches, which learn the AAM without correct answers, we manually label the AAM of reaction data to train the model. While the manual labeling of a large amount of AAM in a large dataset can be an exhaustive and expensive task, we design an active learning framework to manually label only a small fraction of reactions diversely sampled from a large dataset. With these chemist-labeled AAM, we train a graph neural network (GNN) to learn the correct AAM of reaction using both local message passing and long-range attention. For a publicly available USPTO-50K dataset, the model can predict the AAM with 98.5% accuracy only by learning from 2% of the chemist-labeled reactions. More importantly, the AAM of 97% of the reactions in the dataset confidently predicted by LocalMapper shows a 100% prediction accuracy. The same perfect accuracy is observed by testing the model with a diverse out-domain reaction test set. We expect our approach can be used to generate reliable AAM for reaction databases and improve the quality of future ML models relying on AAM. We summarize the important breakthroughs of this paper in three aspects.

1. The proposed knowledge-based uncertainty identification allows the fast chemical-aware verification of ML model predictions, yielding 100% correct AAM for 3,000 randomly sampled confident predictions.
2. The developed model, *LocalMapper*, achieves state-of-the-art AAM prediction accuracy by learning the chemist-verified AAM from high-quality training data curated by human-in-the-loop machine learning. We show a better prediction accuracy compared to the existing ML-based models, RXNMapper[18] and GraphormerMapper[26] by only labeling 2% of the reactions.
3. In an out-of-distribution experiment, LocalMapper shows favorable prediction accuracy over two existing ML-based AAM models, while maintaining 100% accuracy on the confident predictions.

## Results
### The human-in-the-loop machine learning framework
In this work, we propose a graph-based model, *LocalMapper*, to learn the correct AAM through human-in-the-loop machine learning. To train LocalMapper, we manually label the AAM for each reaction to guarantee the correctness of AAMs in the reactions for training the model. Because manual labeling AAM for chemical reactions is intensely time-consuming (in general over one minute per reaction), it is impractical to label a large portion of the reactions in a large dataset. Therefore, we introduce active

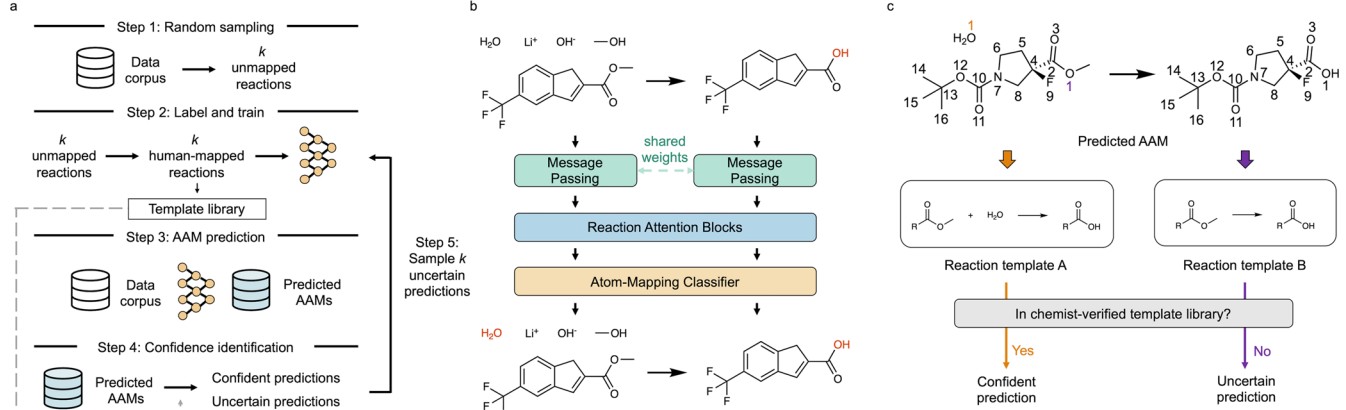

**Fig. 2 | The overall scheme of the human-in-the-loop machine learning for atom-to-atom mapping (AAM) with the proposed model *LocalMapper*. a** The workflow of the proposed active-learning approach. First, we sample $k$ reactions from the entire reaction dataset. After manually labeling the AAM of the sampled reactions, we train the LocalMapper model to learn the correct AAMs. Next, we used the trained LocalMapper to predict the rest of the data and classify the predictions into confident and uncertain predictions via previously verified reaction templates. Then we sample another $k$ reactions from uncertain predictions and label them to train the next generation of models. This cycle is repeated for $n$ iteration until a criterion is reached. **b** The model architecture of LocalMapper. First, both features of reactant and product molecules are updated through message-passing layers. Next, all atoms are free to communicate the information through reaction attention blocks. Finally, a classifier is trained to identify which atom in the reactant corresponds to a specific atom in the product. **c** Knowledge-based confidence identification. After predicting the AAMs of each reaction in the dataset, we determine the prediction confidence by checking whether the reaction template extracted from the mapped reaction has been verified by a human chemist during the manual labeling phase (orange AAM) or not (purple AAM).

learning to label only a small fraction of representative reactions. The overall workflow can be decomposed into the following 5 steps (Fig. 1a), and more details about LocalMapper (Fig. 1b) and prediction confidence (Fig. 1c) are described in the next two subsections.:

1. Random sampling: To initialize the active-learning process, we randomly sample $k$ reactions from the unmapped reaction dataset., where $k$ is an affordable small number for a human expert to label the AAM at one time.

2. Label and train: Next, we manually label the AAM for the sampled $k$ reactions and use these reactions to train the proposed graph-based model *LocalMapper*, structurally similar to the retrosynthesis model LocalRetro[14] and reaction outcome prediction model LocalTransform[32]. Reaction templates extracted from human-mapped reactions are used to update a template library, which will be used for later uncertainty identification.

3. AAM prediction: Next, we use LocalMapper to predict the atom-atom correlation between reactants and products for all the reactions in the dataset. According to the atom-atom correlation predicted by LocalMapper, we generate the AAMs for each reaction following the atom-mapping procedure introduced by Schwaller et al. [18]

4. Confidence identification: For each predicted reaction's AAM, we extract the reaction template to represent its pattern of reactivity. If the extracted reaction template exists in the current template library, the set of AAMs predicted at the reaction is considered a confidence prediction, otherwise an uncertain prediction.

5. Active sampling: For each unique template extracted from uncertain predictions, we sample one reaction starting from the template sharing the most reactions, until $k$ reactions are sampled. These reactions are then labeled by human chemists and used the train the model in the next iteration, repeating step 2.

From the second iteration, we train the model using semi-supervised learning by sampling 100 reactions from the confident predictions from each unique verified reaction template to increase the model's robustness. These sampled reactions are split into the training and validation set by a 9:1 ratio to prevent overfitting.

## LocalMapper

To predict the AAM between the reactant and product in the reaction, we design a graph-based model, called *LocalMapper*, to learn the probability of each atom in the reactant being repositioned to the atom in the product $p(atom_r|atom_p)$. Similar to our previous models for retrosynthesis, LocalRetro[14], and reaction outcome prediction, LocalTransform[32], we use the graph to represent molecules, with atoms as nodes and bonds as edges, and learn the AAM by both local and global features of the atoms in the reactions by message passing neural networks[33] and attention mechanism[34] (Fig. 2b).

First, we encode the local chemical environment of each atom using 3 message-passing layers[33] and update the atom features in the product by atom features from the reactants through 3 multi-head cross-attention blocks[34]. After the features of each atom between reactants and products are sufficiently communicated, we calculate the AAM correlation between product and reactant by a single-head attention block. After normalizing the attention scores with the Softmax function, the probability of each atom in the reactant being the same atom of each atom in the product is estimated. Following the atom-mapping procedure introduced in RXNMapper[18], we use the resulting probability to identify the AAM from product to reactant from the highest probability to the lowest probability. In the example shown in Fig. 2b, oxygen from the water molecule at the reactant side is identified as the source of the oxygen on ketone in the product molecule. The mathematical details of each layer and pseudocode of LocalMapper can be found in the Method Section.

## Knowledge-based prediction confidence

Accessing the prediction confidence is one of the most important features of ML models, which informs the use of whether the model's prediction is reliable or not. Sampling and labeling uncertain predictions to train the model is usually referred to as active learning, which can efficiently explore the necessary data to label for the model to further learn. Popular methods of quantifying the prediction confidence include Monte Carlo dropout[35], bootstrapping[36], and multiplying the prediction probabilities[18,37]. Despite a positive correlation between accuracy and uncertainty using these approaches, none of these approaches use in-domain knowledge but solely depend on the model parameters.

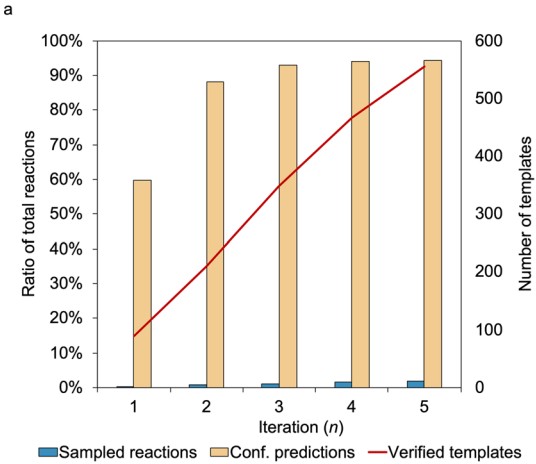

**Fig. 3 | The results of active-learning for training LocalMapper in this work. a** The statistics of active learning from iteration $n=1$ to 5. **b** A nucleophilic methylation reaction sampled from active-learning at $n=1$. **c** A nucleophilic acyl substitution reaction sampled from active-learning at $n=3$. **d** A thiazole synthesis reaction was sampled from active-learning at $n=5$. Molecules without AAMs are removed in the depiction for reading clarity. Source data are provided as a Source data file.

Here, we introduce a knowledge-based approach to identify prediction confidence by examining the presence of a reaction template derived from the predicted AAM in the chemist-verified template library. Since the reaction mechanism of a chemical reaction is determined by itself, we assume there exists only one correct chemically reasonable reaction template that can be derived from the correctly predicted reaction AAM. On the other hand, reactions with incorrectly predicted AAMs would not give chemically reasonable templates. Therefore, we define the prediction as confident if the reaction template derived from the predicted AAM has already been identified and verified by a human expert during the manual reaction labeling process, otherwise classified as uncertain predictions. Hence, the uncertain predictions are either wrong or correct but have not been validated by human experts yet. These uncertain predictions can be sampled and confirmed by human experts in the active learning process. The example illustrated in Fig. 1c shows that only when the AAM of the given reaction is correctly predicted would it yield a chemically reasonable reaction template (template A) and be identified as a confident prediction, otherwise uncertain (template B). We use an extended version of the local reaction template, extended-local reaction template (ELRT), to represent the reactions in this work. See the "Methods" section and Supplementary Section 1 for more details about ELRT.

## Results of active-learning

In our experiments, we perform active-learning to train LocalMapper on the USPTO-50K dataset, containing 49,996 reactions curated by Schneider et al. [38], by sampling 200 reactions at each active-learning iteration and repeating for 5 iterations ($k=200$, $n=5$). The number of reaction templates and the prediction coverage, the percentage of predicted AAM yielding reaction templates existing in the template library, are shown in Fig. 3a. At the first iteration, 200 reactions were randomly sampled from all the reactions. The AAMs of these reactions yield 90 unique reaction templates and cover 59.7% of the total reactions. As the iteration increases, the number of unique reaction templates steadily increases (209, 348, 467, 555 for $n=2, 3, 4, 5$), while the increment of prediction coverage delays over iterations (88.1%, 92.9%, 93.9%, 95.0% for $n=2, 3, 4, 5$), meaning the less popular reaction templates were sampled in later iterations. This is a great example of how active-learning was used to prioritize the sampled reactions to maximize the sample efficiency.

Next, we show examples of sampled reactions during active-learning at $n=1, 3, 5$ in Fig. 3b–d. At $n=1$, the simple and popular reactions such as substitution reactions and redox reactions were sampled. The reaction shown in Fig. 3b is one of the nucleophilic acyl substitution reactions, accounting for 11.0% of the total reactions in the dataset. At $n=3$, more organometallic reactions such as Gridnard reactions and Stille coupling were sampled. The reaction shown in Fig. 3c is a nucleophilic methylation with methyllithium as the methylating reagent. At $n=5$, several ring-forming intramolecular reactions were sampled. The reaction shown in Fig. 3d is a thiazole synthesis reaction from imine and primary thioamides.

## AAM evaluation

To assess the prediction accuracy of LocalMapper, we conducted a comparative analysis with two unsupervised learning-based models: RXNMapper[18] and GraphormerMapper[26]. To ensure a fair model comparison, we evaluate these models without the use of AAM fixer[39], which automatically fixes the known incorrect AAMs to correct AAMs after predictions. We implemented these models using their publicly available software on GitHub. Before evaluating the AAM models., we filter out the reactions from the USPTO-50K dataset if they include invalid product mapping and confusing reagents as previously reported by Schwaller et al.[18]. The former criteria filters out reactions with a product showing repeating atom-mapping or atoms without atom-mapping, while the latter criteria filters out reactions having reactants structurally similar (Tanimoto similarity ≥ 0.5) to the product but not participating in the reaction. Following these criteria, 1166 reactions were excluded, leaving 48,830 reactions for AAM evaluation. More definitions and examples of problematic reactions can be found in Supplementary Section 2.

Given that the USPTO-50K dataset was known to have potentially incorrect AAMs[10,16,18], we report three different accuracy metrics in this article. The first metric assumes the AAM recorded in the dataset as ground truth (referred to as "dataset accuracy" or $\text{Accuracy}_{\text{overall}}^{\text{dataset}}$). The second metric involved manually checking the 3000 sampled confident predictions generated by both RXNMapper or LocalMapper (referred to as "manual checked accuracy", $\text{Accuracy}_{\text{conf.}}^{\text{manual}}$). Lastly, we introduced a "calibrated accuracy" metric ($\text{Accuracy}_{\text{overall}}^{\text{calibrated}}$, Eq. 1) by combining the results from the dataset accuracy and the manually checked accuracy.

$$\text{Accuracy}_{\text{overall}}^{\text{calibrated}} = \text{Accuracy}_{\text{unconf.}}^{\text{dataset}} \times \text{Ratio}_{\text{unconf.}} + \text{Accuracy}_{\text{conf.}}^{\text{manual}} \times \text{Ratio}_{\text{conf.}}$$

$$(1)$$

**Table 1 | Atom-to-atom mapping (AMM) results of RXNMapper, GraphormerMapper, and LocalMapper on the USPTO-5OK dataset before and after manually checking the reaction AAMs**

| Model | Dataset accuracy | | Manual checked accuracy | | Calibrated accuracy |
|---|---|---|---|---|---|
| | All predictions | Conf. predictions | Conf. predictions | Conf. ratio | All predictions |
| RXNMapper[18] | **98.1%** | **99.7%** | 93.6%[a] | 30.9% | 96.2% |
| GraphormerMapper[26] | 92.8% | 94.6%* | 94.8%[a] | / | / |
| LocalMapper (this work) | 91.5% | 92.8% | **100%[a]** | **97.0%** | **98.5%** |

The best accuracy and ratio are highlighted in bold font.

[a]Accuracy evaluated on the 3000 reactions sampled from the reactions confidently predicted by both RXNMapper and LocalMapper.

where the accuracy of unconfident prediction is estimated by

$$\text{Accuracy}_{\text{unconf.}}^{\text{dataset}} = \frac{\left(\text{Accuracy}_{\text{overall}}^{\text{dataset}} - \text{Accuracy}_{\text{conf.}}^{\text{dataset}} \times \text{Ratio}_{\text{conf.}}\right)}{\text{Ratio}_{\text{unconf.}}} \quad (2)$$

Because RXNMapper also gives a confidence score for each prediction, which shows a positive correlation with the prediction accuracy[18], we binarize the confident score of RXNMapper by its prediction confidence score of 0.9 (according to the best performing results shown in the Supplementary Material of ref. [18]) to facilitate the comparison with the confident predictions generated by LocalMapper. Note that GraphomerMapper does not generate a confidence score with its prediction; therefore, we did not assess the accuracy of the confident prediction of this model. The accuracy of AAM predictions is calculated by comparing the condensed graph of reaction (CGR) between the model's prediction and the ground truth using CGRtools toolkit[2] following previous works[26,39] to ensure that equivalent but different AAMs between the ground truth and model predictions did not lead to underestimations of prediction accuracy.

The results of AAM compared with RXNMapper and GraphormerMapper on the USPTO-5OK dataset are shown in Table 1. Before we conducted manual checks to assess the correctness of the dataset's AAM, RXNMapper exhibited an impressive overall accuracy of 98.1% on the 48,830 reactions in the USPTO-5OK dataset. In comparison, GraphormerMapper demonstrated a commendable overall accuracy of 92.8%, based on the dataset AAM. Moreover, within the prediction generated by RXNMapper, 30.4% of the confident predictions (i.e., with a confidence score exceeding 0.9) show a nearly perfect accuracy at 99.7%. In contrast, LocalMapper yields a high ratio of confident predictions at 97% but only exhibits a 91.5% overall prediction accuracy, noticeably lower than RXNMapper's accuracy and slightly behind that of GraphormerMapper. For these confident predictions from LocalMapper, the calculated accuracy was 92.8% based on the dataset AAM.

To investigate the incorrectly predicted confident predictions from RXNMapper and LocalMapper, we randomly sampled 3000 reactions from a pool of 14,422 reactions confidently predicted by both RXNMapper and LocalMapper. After manually checking these predicted AAMs, we found all the confident predictions from LocalMapper are indeed correct, but they have been incorrectly mapped in the original dataset. In particular, within 3000 randomly sampled reactions, 6.6% of them were ester hydrolysis reactions, and they were all correctly predicted by LocalMapper but incorrectly predicted by RXNMapper. It is worth highlighting that these reactions were initially misaligned in the dataset's AAM, matching RXNMapper's AAM predictions, further indicating the potential for overestimating RXNMapper's prediction accuracy and underestimating LocalMapper's performance based on the dataset's AAM. To address this discrepancy, we recomputed the calibrated accuracy using Eq. 2, which aims to reflect the actual prediction accuracy more accurately. Consequently, the calibrated accuracy showed LocalMapper achieving a higher accuracy rate at 98.5% compared to RXNMapper's 96.2%. Moreover, it is essential to emphasize that 97% of confidently

predicted AAMs generated by LocalMapper are highly likely to exhibit perfect accuracy.

To qualitatively compare and gain insights into the differences between LocalMapper and the second-best performing model, RXNMapper, we conducted a detailed analysis of AAMs between the dataset, RXNMapper, and LocalMapper, as visually represented in Fig. 4a through a Venn diagram. Among the reactions within the dataset, 90.5% of reactions were found to have equivalent AAMs. For the remaining 9.5% of reactions where the predicted AAMs differed, RXNMapper shared 7.6% of equivalent AAMs with the dataset, while LocalMapper exhibited lower overlap, sharing only 1% and 0.8% with the dataset and RXNMapper, respectively. These statistics provide insight into the "low accuracy" of LocalMapper when assuming the dataset's AAMs as ground truth. In Fig. 4b, c, we illustrate two examples of unique AAM predictions generated by LocalMapper. These examples represent ester hydrolysis and acetal hydrolysis reactions, respectively. In Fig. 4b, LocalMapper correctly mapped the highlighted oxygen atom in the product (number 16) to water in ester hydrolysis reactions, whereas RXNMapper and the dataset suggested that the oxygen originated from the leaving group. In Fig. 4c, LocalMapper accurately mapped the highlighted oxygen atom in the product (number 16) to water, whereas RXNMapper and the dataset consistently misattributed the oxygen atom to the acetal oxygen.

Further analysis of the reaction templates of the 3,308 confident predictions generated by LocalMapper, which differed from the dataset's AAM, revealed interesting insights. Among these unique predictions, 81.7% were ester hydrolysis reactions (as shown in Fig. 4b), 8.1% were esterification reactions (Fig. 4e), 3.5% were acetal hydrolysis reactions (Fig. 4c), and 0.3% were Mitsunobu reactions (Fig. 4f). These AAMs were all correctly predicted by LocalMapper but incorrectly mapped in the original dataset.

Next, we examine the generalizability of LocalMapper on the golden dataset compiled by Lin et al.[39] including 1851 reactions after standardizing (fixing invalid valences or radicals) and manually mapping the reactions collected from Jaworski et al.[17] and popular reactions from the USPTO collections[40]. We found there are 90 unbalanced reactions, 2 repeated reactions, and 1 reaction without product in the golden dataset. Consequently, we evaluated the models on the remaining 1758 reactions. Examples of unbalanced reactions can be found in Supplementary Section 3.

Since this dataset mixes the sources from the original literature, it is hard to analyze the model performance on different reaction sources. Therefore, we compared the reactions recorded in the golden dataset with the categorized reaction sets compiled by Jaworski et al.[17] and extracted 491 reactions from the golden dataset, including 256 USPTO reactions[38], 173 typical reactions from the Organic Synthesis collection[41], and 62 mechanistically complex reactions from various literature sources[42,43]. To enhance LocalMapper's ability to confidently predict a wider spectrum of organic reactions, we conducted further training of the model for 2 additional iterations on the USPTO-FULL dataset[18,44] (containing 1,065,119 reactions) with sampling 500 reactions at each iteration ($k = 500$, $n = 2$). All the reactions in this test set were excluded from the training set of LocalMapper.

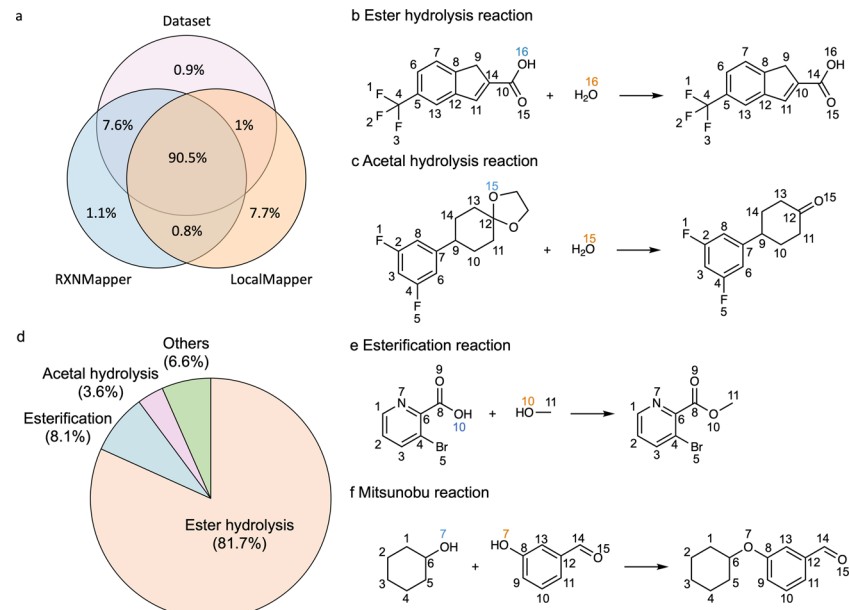

**Fig. 4 | Overall results of LocalMapper predicting atom-to-atom mappings (AAMs) on the USPTO-50K dataset. a** The Venn diagram of AAMs in the dataset and predicted by RXNMapper and LocalMapper. **b, c** Two examples of the AAMs predicted by LocalMapper and other models for **b** ester hydrolysis and **c** acetal hydrolysis reactions. The different AAMs predicted by LocalMapper and other methods are highlighted in orange (LocalMapper) and blue (Dataset and RXNMapper). **d** The ratio reaction types among the"incorrect confident predictions" generated by LocalMapper on the USPTO-50K dataset. **e, f** Two examples of the "incorrect confident predictions" by LocalMapper for **e** esterification and **f** Mitsunobu reaction. The unchanged AAMs are annotated in black, the wrong AAMs in the dataset and RXNMapper are highlighted in blue, and the AAMs predicted by LocalMapper are highlighted in orange. Molecules without AAMs are removed in the depiction for reading clarity. Source data are provided as a Source data file.

**Table 2 | Atom-to-atom mapping (AMM) results of RXNMapper, GraphormerMapper, and LocalMapper on manual-mapped reactions examined on four different sources**

| Model | Golden[39] | USPTO[38] | Typical[41] | Complex[42,43] |
|---|---|---|---|---|
| | **Accuracy of all predictions** | | | |
| RXNMapper[18] | 86.5% | 89.5% | 91.3% | 59.7% |
| GraphormerMapper[26] | 82.7% | 93.8% | 87.9% | 66.1% |
| LocalMapper (this work) | **89.8%** | **99.2%** | **93.6%** | **69.4%** |
| **Accuracy (ratio) of confident predictions** | | | | |
| RXNMapper[18] | 95.1% (19.7%) | 90.2% (23.8%) | 95.7% (13.3%) | 50.0% (**12.9%**) |
| LocalMapper (this work) | **100%** (**53.3%**) | **100%** (**79.7%**) | **100%** (**42.8%**) | **100%** (6.5%) |

The highest accuracy and ratio are highlighted in bold font.

The results compared with RXNMapper[18] and GraphormerMapper[26] are shown in Table 2. When evaluating the models on the golden dataset, we found there are 8 reaction AAMs confidently predicted from LocalMapper but different from ground truth. We found these reactions are either wrongly mapped in the golden dataset (7 reactions) or selective reactions (1 reaction), in which multiple AAMs are acceptable. Therefore, we show the prediction results after calibrating the accuracy after manual checking in Table 2. The different AAMs and the original results following ground truth AAMs can be found in Supplementary Section 4.

When predicting on the full golden dataset, irrespective of reaction sources, LocalMapper achieves an impressive 89.8% prediction accuracy, surpassing RXNMapper by 3.3% and GraphormerMapper by 7.1%. Focusing on the 256 USPTO reactions, LocalMapper excels with a remarkable 99.2% prediction accuracy, outperforming GraphormerMapper by 5.4% and RXNMapper by 9.7%. For typical reactions, LocalMapper achieves a prediction accuracy of 93.6%, exceeding the other two models by margins of 2.3% and 5.7%. In the case of complex reactions, LocalMapper secures the second-highest prediction accuracy

at 69.4%, slightly surpassing GraphormerMapper (66.1%) and greatly higher than RXNMapper (59.7%). Importantly, the ratio of confident predictions across different datasets exhibits variations, yet the prediction accuracy of these confident predictions consistently remains at 100% for all four examined datasets. In contrast, while RXNMapper demonstrates over 90% confident prediction accuracy for the golden dataset, USPTO reactions, and typical reactions, it only achieves a 50% confident prediction accuracy for complex reactions despite the double ratio of confident predictions compared to LocalMapper.

It's worth noting that the confident prediction ratio of Local-Mapper tends to decrease when applied to reactions that differ more significantly from the training reactions, i.e., USPTO reactions. This trend is evident in the prediction accuracy of LocalMapper, which decreases from 99.2% for USPTO reactions to 93.6% for typical reactions, and further to 69.4% for complex reactions. These findings underscore an essential insight from LocalMapper: not only are its confident predictions highly reliable, but the overall prediction accuracy for a set of reactions can be estimated based on the ratio of confident reactions.

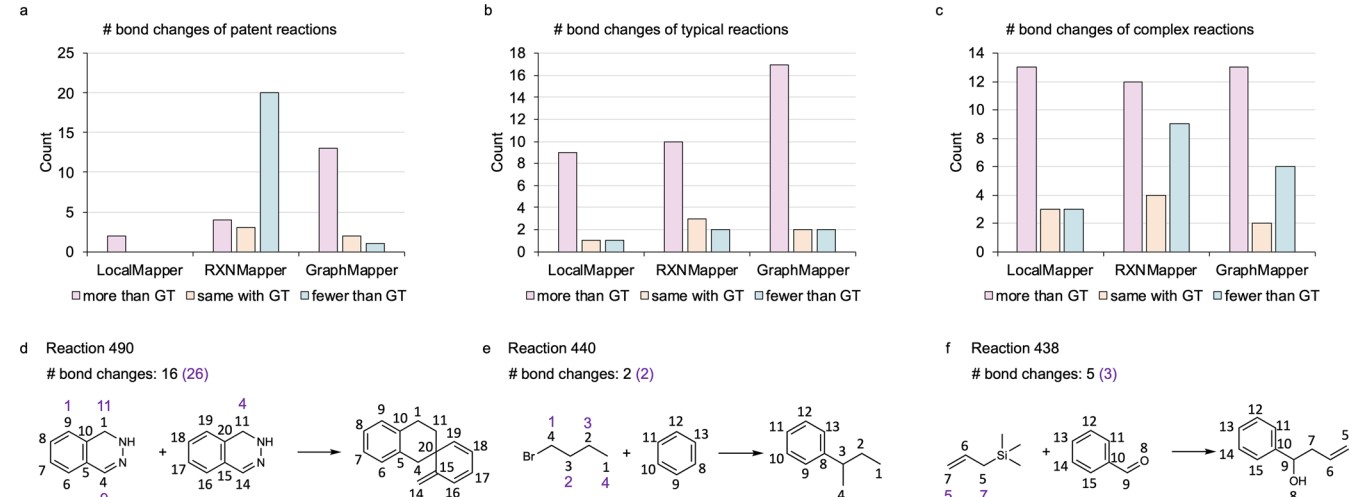

**Fig. 5 | Comparative analysis of the number of bond changes in reaction AAMs.** The figure illustrates the number of bond changes derived from the atom-to-atom mappings (AAMs) of **a** patent, **b** typical, and **c** complex reactions, incorrectly predicted by LocalMapper, RXNMapper, and GraphormerMapper (GraphMapper) compared with the ground truth (GT) AAMs. Additionally, examples of complex reactions with AAM predictions from LocalMapper are presented, showcasing instances where the predicted AAM yields **d** more, **e** the same, or **f** fewer number of bond changes compared to the ground truth AAMs. Ground truth AAMs and corresponding bond change counts for the example reactions are annotated in black, while those generated by LocalMapper are highlighted in purple. Source data are provided as a Source data file.

In Fig. 5a–c, we conduct a detailed analysis of the number of bond changes derived from the AAMs (according to their CGRs) of reactions that were inaccurately predicted by LocalMapper, RXNMapper, and GraphormerMapper, respectively. Generally, the majority of incorrectly predicted AAMs across all three models and various reaction sources result in an increased number of bond changes compared to the corresponding ground truth AAMs. Notably, RXNMapper stands out for producing a substantial number of incorrectly predicted AAMs that result in a decreased number of bond changes in patent reactions, primarily involving ester hydrolysis reactions. To illustrate the impact of such predictions, we present an example in Fig. 5d, wherein even a small number (4) of incorrectly predicted AAMs can result in a significantly higher count (10) of bond changes compared to the correct AAMs. However, the examples depicted in Fig. 5e, d underscore that predicted AAMs, showing either the same or fewer bond changes, do not consistently align with the ground truth AAMs, especially in the context of complex reactions.

## Discussions

The key distinction between LocalMapper and the other two existing ML-based models lies in the use of chemist-labeled data during training. While training without manual mapping may offer computational efficiency, it can lead to unforeseen systematic errors, such as those seen in AAM predictions for simple reactions in Fig. 4. This emphasizes the importance of meticulous data labeling, despite its time and expertise demands. Moreover, manual labeling yields valuable chemical rules that can be leveraged for robust knowledge-based prediction confidence identification, which contributes to the 100% confident prediction accuracy of LocalMapper.

It is vital to distinguish between LocalMapper's knowledge-based prediction confidence and the AAM fixer employed in GraphormerMapper[26]. While both methods leverage the insights of chemists to enhance prediction accuracy, they diverge significantly in their operational mechanisms. The AAM fixer directly rectifies the model's AAM predictions but does not enhance the model itself. It relies on manual heuristics to correct known inaccuracies, making it challenging to scale up without extensive experimental adjustments. In contrast, knowledge-based prediction confidence empowers human chemists to label uncertain predictions, thereby facilitating model improvement through active learning. This approach is data-driven and easily scalable by expanding active learning iterations. We have also considered a model-free approach involving the direct application of all known reaction templates to the reactants for obtaining the reaction AAM by matching the known product. While this method enhances the robustness of AAM prediction, mapping on the USPTO-50k dataset requires approximately 10 times longer, taking 6 h to complete compared to the 35 min required by LocalMapper.

Although we labeled only 47.3 K reactions (97%) in the USPTO-50K dataset after manually annotating 1000 reactions in this paper, it is remarkable that the same model can confidently label 544.5 K reactions (51.1%) in the full USPTO dataset. This represents a substantial increase in the labeling efficiency achieved through manual annotation. Furthermore, with two additional active learning iterations, this number grows to 712.6 K reactions (66.9%). However, it is important to note that LocalMapper tends to yield a significantly lower ratio of confident predictions when applied to entirely distinct reaction datasets, such as quantum-mechanical reactions[45] and enzymatic reactions[46]. These reactions may follow entirely different reaction mechanisms (unimolecular one-to-many reactions and enzyme-catalyzed reactions, respectively) that were never encountered during active learning on organic reaction datasets. For such cases, we recommend engaging domain-specific chemists to undertake active learning iterations to adapt the model effectively before using it for large-scale AAM tasks.

In summary, we propose a graph-based ML model, LocalMapper, to precisely identify the AAM for large reaction datasets via human-in-the-loop machine learning. By manually labeling a small amount of reaction data with expert knowledge, we train an human-in-the-loop ML model to precisely and automatically label a large number of reactions sharing similar reaction rules. The proposed knowledge-based active sampling enables the human expert to only label the AAM of 2% of reactions that include the reaction templates of 97.0% of reactions in the entire dataset. We show an overall 98.5% AAM prediction accuracy, with 100% accuracy for confident predictions on a widely used USPTO-50k dataset, and a similar result is also observed in a diverse out-of-distribution test set. We expect the proposed Local-Mapper can be used to provide precise reaction AAMs for future downstream reaction prediction models and benefit the chemistry

community to learn more statistical insights into the reaction dataset. The trained LocalMapper model and the generated AAMs on USPTO-50K and USPTO-FULL datasets introduced in this paper are available at https://github.com/snu-micc/LocalMapper.

## Methods

### Extended-local reaction template (ELRT)
After the AAM is identified from LocalMapper, we extract the reaction template from the mapped reaction to categorize the reaction type of the given reaction. As a popular reaction template extraction tool, RDChiral[47] was developed to extract the reaction template considering atom neighbors, special groups, and stereochemistry and has been used in many retrosynthesis prediction models. However, the reaction template extracted by RDChiral was considered to be too specific, leading to low generalizability of reactions with the same reaction type (four reactions per template on average, extracted from USPTO-50K). Therefore, Chen and Jung[14] modified the reaction template to only focus on local changes, which significantly improved the template generalizability to 76 reactions per template on average. Despite the enhanced generalizability of the local reaction template, important functional groups, such as acetal, carbonyl group, and nitrile, need to be included to make the reaction template more chemically understandable for the present purpose of AAM. Therefore, we extend the local reaction template by including important functional groups and denote it as extended-local reaction template (ELRT). More examples and the full set of functional groups included in the ELRT can be found in Supplementary Section 1.

Due to the absence of essential reagent and catalyst information in many reactions within the USPTO-50K dataset, we do not incorporate reagent and catalyst details into the reaction templates. For instance, we observed that at least 1166 (49.4%) out of 2362 Suzuki coupling reactions lack Pd catalyst, 520 (34.6%) out of 1500 nitro reduction reactions do not feature a reduction agent, and 170 (39.4%) out of 431 Mitsunobu reactions do not include diethyl azodicarboxylate (DEAD) or diisopropyl azodicarboxylate (DIAD). As a result, we make the simplifying assumption that common and necessary reagents or catalysts are present in the reactions during template extraction.

### Molecular graph
The inputs of LocalMapper are the graphs of reactants and products of the target reaction We represent the reactant graph as $G_r = (V_r, E_r)$ and the product graph as $G_p = (V_p, E_p)$, where $V$ (vertices) denotes atoms and $E$ (edges) denotes bonds. The initial atom and bond features are the same as the ones used in LocalRetro[14] and LocalTransform[32], available in Supplementary Section 6. Both graphs are built using the DGL-LifeSci[48] Python package. The features of each atom in the reactants are denoted as $h_{r,u}$ (for atom $u$) and the features of each bond in the reactants are denoted as $h_{r,uv}$ (for the bond between atom $u$ and atom $v$). Similarly, the features of each atom and bond in the products are denoted as $h_{p,u}$ and $h_{p,uv}$.

### Message massing neural network (MPNN)
To encode the surrounding environmental information for each atom, we used a message-passing neural network (MPNN)[33,49] to update the atom features for 3 iterations. We denote the message passing function by MPNN($\cdot$), which update the atomic features $h_u$ of atom $u$ by its neighbor atoms $\{v\}$ and bonds $\{uv\}$ in the molecule (Eqs. 3 and 4).

$$h_{r,u}^{t+1} = \text{MPNN}\left(h_{r,u}, \{h_v\}_{v \in V_r}, \{h_{uv}\}_{uv \in E_r}\right) \quad (3)$$

$$h_{p,u}^{t+1} = \text{MPNN}\left(h_{p,u}, \{h_v\}_{v \in V_p}, \{h_{uv}\}_{uv \in E_p}\right) \quad (4)$$

### Reaction attention
After encoding the local chemical environment of each atom in the individual molecule, we enable the atoms in the product to refine their features by looking at the atoms in the reactants through multi-head attention blocks[34]. In particular, we used multi-head attention MultiHeadAtt($\cdot$) between the atoms in the products and reactants:

$$\text{MultiHeadAtt}\left(h_{p,u}, \{h_v\}_{v \in G_r}\right) = \text{Concat}\Big(\text{head}_2\left(h_{p,u}, \{h_v\}_{v \in G_r}\right),$$
$$\text{head}_2\left(h_{p,u}, \{h_v\}_{v \in G_r}\right), \ldots, \text{head}_n\left(h_{p,u}, \{h_v\}_{v \in G_r}\right)\Big) \quad (5)$$

where Concat($\cdot$) is the concatenation operation between each attention head.

The output of each attention head is the updated atoms features according to attention score $e_{u,v}$ and value $V_{n,v}$

$$\text{head}_n(h_u, \{h_v\}) = \sum \text{Softmax}(e_{u,v})V_{n,v} \quad (6)$$

where attention score $e_{u,v}$ is computed by the query $Q$, key $K$, and value $V$ of each atom features, which are calculated by the linear layers in each attention head, and normalized by the hidden dimension $d$ and the number of attention head $n$:

$$Q_u = w_Q h_u \quad (7)$$

$$K_v = w_K h_v \quad (8)$$

$$V_v = w_V h_v \quad (9)$$

$$e_{u,v} = \frac{Q_u(K_v)^T}{\sqrt{d/n}} \quad (10)$$

In our experiment, we used 3 reaction attention blocks with 8 attention heads in each attention block. The dropout rate in the multi-head self-attention layer was set to 0.1. Gated transformation, skipped-connection, and layer normalization were applied after the attention mechanism and followed by standard feed-forward neural networks

$$h_{p,u}^{t+1} = h_{p,u}^t + w_f\left(\text{Sigmoid}\left(w_g m_{p,u}^t + b_g\right)\right) + b_f \quad (11)$$

where $w_f$ and $b_f$ are the weights and biases of feed-forward neural networks, $w_f$ and $b_f$ are the weights and biases of gated transformation, and $m_{p,u}^t$ is the message of atom $u$ in the product $t$ obtained from the multi-head attention block at step $t$.

### Atom-mapping classifier
Finally, the AAM score between atom $u_p$ in the products and atom $u_r$ in reactants $p(u_r|u_p)$ was computed by another single-head attention block as an atom-mapping classifier:

$$p(u_r|u_p) = \text{Classifier}(h_{r,u}|h_{p,u}) = \text{Softmax}\left(\frac{Q_{r,u}(K_{p,u})^T}{\sqrt{d}}\right) \quad (12)$$

### Training objectives
The designed LocalMapper model is trained to optimize the AAM score $p(u_r|u_p)$ between each pair of corresponding atoms in the products and reactants through cross-entropy losses. Let $(u_{r,i}, u_{p,i})$ be the pair of atoms in the products and reactants sharing the same atom-number $i$, the objective of AAM to train the model parameter $\theta$

is

$$\mathcal{L}_{\mathrm{AAM}} = \max_{\boldsymbol{\theta}} \mathbb{E}\left[\log(p_{\boldsymbol{\theta}}(u_{r,i}|u_{p,i}))\right] \qquad (13)$$

## Training hyperparameters and pseudocode

All the chemical operations are done using the RDKit[50] python package. We used Pytorch[51] and DGL-LifeSci[48] for neural network training and testing. We train our model for 100 epochs with batch size 16 by Adam optimizer[52] with $10^{-6}$ weight decay and set the initial learning rate to $10^{-3}$. The learning rate is reduced by a factor of 0.5 after when the validation loss does not decrease after a training epoch. Model gradients are clipped at a maximum norm of 20. Training LocalMapper takes around 4 h for the USPTO-50K dataset (at fifth iteration) and 6 h for the full USPTO dataset (at second iteration), while the inference takes 35 min for the former and 14 h for the latter datasets. The pseudocode of training LocalMapper is given in algorithm 1.

**Algorithm 1**. The pseudocode of training *LocalMapper*.

```
Input    :
Reactant graph G_r(V_r, E_r); reactant features {h_{r,u} ∈ V_r, h_{r,uv} ∈ E_r};
product graph G_p(V_p, E_p); product features {h_{p,u} ∈ V_p, h_{p,uv} ∈ E_p};
AAM label y

Initialize:
Model parameters θ

1  for epoch in 1, N do
2  │  h_{r,u} ← MPNN(h_{r,u}, {h_{r,uv}})
3  │  h_{p,u} ← MPNN(h_{p,u}, {h_{p,uv}})
4  │  h_{p,u} ← MultiHeadAtt(h_{p,u}, {h_{r,u}})
5  │  p(u_r|u_p) ← Classifier(h_{p,u}, h_{r,u})
6  │  loss ← L_{AAM}(p(u_r|u_p), y)
7  │  update θ
8  end
```

## Reporting summary

Further information on research design is available in the Nature Portfolio Reporting Summary linked to this article.

## Data availability

The AAMs for 3000 sampled reactions on the USPTO-50K dataset and out-of-distribution reactions predicted by the three evaluated ML models can be found at https://github.com/snu-micc/LocalMapper[53]. The USPTO-50K and USPTO-full datasets remapped by LocalMapper generated in this study have been deposited in Figshare (https://doi.org/10.6084/m9.figshare.25046471.v1[54]). Source data are provided with this paper.

## Code availability

The code for LocalMapper described in this manuscript is publicly available at https://github.com/snu-micc/LocalMapper.

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

## Acknowledgements

This work was supported by the Digital Research Innovation Institution Program funded by NRF Korea (RS-2023-00283902, Y.J.), Technology Innovation Program funded by MOTIE Korea (20015850, Y.J., S.C., S.A.), SRC Center for Electron Transfer (2021R1A5A1030054, Y.J., S.A.) funded by NRF Korea, and AI Graduate School Program of SNU funded by IITP Korea (2021-0-01343, Y.J.).

## Author contributions

S.C. designed the methods, performed the computational experiments and analyses, and wrote the initial draft of the manuscript. S.A. assisted the computational experiments. R.B. assisted to evaluate the mechanism of complex reactions. Y.J. discussed the results, edited the manuscript, and supervised the project.

## Competing interests

The authors declare no competing interests.
