## [Peer Review File · Nature Communications]

REVIEWER COMMENTS

Reviewer #1 (Remarks to the Author):

Jung and colleagues have developed a novel algorithm for atom-to-atom mapping (AAM) in reactions, a fundamental component of reaction analytics. This algorithm utilizes deep learning techniques, trained on manually mapped data through an active learning approach. The authors provided a comprehensive overview of the significance of the AAM task, along with a thorough examination of some existing methods and tools in their article.

However, the manuscript contains some drawbacks:

- Doesn't cover benchmarking done before (<https://doi.org/10.1002/minf.202100138>) on a much larger test set. The performance of the algorithm should be reevaluated on a set from the given article.
- The article above also introduces an AAM comparison algorithm that does not directly involve the comparison of labeled reaction SMILES strings. While the algorithm's specifics are somewhat challenging to discern from the text, it appears to share similarities.
- Template-based noise reduction and ELRT look like the method proposed in (<https://doi.org/10.1021/acs.jcim.2c00344>). If yes – what is the difference? If not – the same question.
- The authors conduct a performance comparison with RXNMapper; however, the article above showcased superior performance compared to RXNMapper. It might be more appropriate to compare against both existing mappers or solely against Graphormer.
- USPTO267 provided by authors showed 98% accuracy on chytorch-rxnmap installed from PyPI.
- The confidence derived from RXNMapper represents an average attention weight and depends on the number of tokens. Consequently, it necessitates experimentation to ascertain its suitability for intended usage.
- It's not clear how the attention block and classifier head are implemented.

A major review of the manuscript should be done. Benchmarking should be done at least on the dataset from the article above. A comparison should be done to the current state-of-the-art mapper or better to a set of mappers. Technical details of the attention and classifier architecture should be clear. Scheme or pseudocode should be provided.

Reviewer #2 (Remarks to the Author):

Jung et al describe a new atom mapping framework based on active learning and manually (re-)labelled data by human experts. They claim a 100% prediction accuracy of their trained mapper after several rounds of active learning, at a confidence rate of 97% for USPTO-50k. The paper is well written, and addresses a very important open question in the field, namely how to obtain high-quality mappings for

reaction datasets. The contribution of the authors advances the field in terms of accuracy and obtaining a measure of confidence for a prediction based on expert level knowledge (the existence of a hand-curated template from the active learning loop). Clearly, this framework was a lot of work to build both regarding the machine learning aspects, as well as the human labeling effort, the analysis of the results and correction of mappings in USPTO-50K, and I deeply appreciate this effort. The code is open-source, and I was able to install and use it easily (with some adaptations for CPUs). The study is reproducible given the information in the manuscript and the code online, and the methodology is adequate. So overall, this is a great piece of work, apart from one major concern: The scope of LocalMapper caused by its use of only USPTO-50K:

- I am concerned about the limited scope of the USPTO-50K dataset underlying the study, as it only comprises a very limited set of chemical reactions, as often shown or highlighted in literature. The authors should demonstrate their work on larger and more diverse datasets, too. Can the mapper be used as-is on larger datasets, such as full USPTO (available free-of-charge), Pistachio, Reaxsys, to name just a few? Or would it require more active learning loops? How much manual labelling is estimated to cover those datasets? Will LocalMapper be actively developed and ready to use on larger datasets in the near future?

- The accuracy of the confident predictions of LocalMapper are indeed very good. When I tested the code on different datasets (following the authors suggestion to compare templates instead of raw mappings), the confident predictions were right in all cases, and revealed a few errors in eg. USPTO-1k-TPL. Thus, the 100% accuracy seems to hold indeed, which is great! However, the ratio of confident predictions gets lower, the more the reactions differ from what is found in USPTO-50k. For example, on a subset of USPTO-1k-TPL <https://doi.org/10.1038/s42256-020-00284-w> (450K USPTO reactions with templates that occur more than 100 times), I find 9% not confident but correct mappings, and 4% not confident and incorrect mappings. So, although still USPTO, the confidence ratio is now only 87%. For those that are not confident and incorrect, LocalMapper provides VERY (like ripping apart molecules) wrong mappings in some cases, so it seems to not generalize well to reactions beyond the known templates in some cases. For quantum-mechanical datasets of reactions (where the ground-truth mapping is known), I find for a subset of the wb97xD3 dataset by Grambow et al <https://doi.org/10.1038/s41597-020-0460-4> (12K elementary reactions), 58% not confident but correct templates, and 38% not confident and incorrect mappings. Thus, the confidence ratio is now only 4%. Sure, the accuracy of confident mappings is 100%, but if 96% of my dataset results in unconfident predictions, this is not very helpful. Further moving away from USPTO-50k, when I test on enzymatic reactions, namely MetaCyc (where MetaCyc mappings do have some flaws, but most of them are correct), I find 40% not confident but correct, and 57% not confident and incorrect predictions, with a confidence ratio of 3%. This highlights the very limited scope of USPTO-50k that the authors rely on, with a sharp drop in the confidence ratio, as well as the ratio of correct templates for unconfident predictions. I think the authors need to put more work in either making LocalMapper more general, or changing the manuscript to highlight that the given confidence ratio is only indicative for the narrow set of USPTO-50K, and people need to retrain it on their dataset (making its use less attractive possibly).

- The above issue is incredibly hard to tackle: With a human in the loop, the covered reactions will necessarily be limited. Without human intervention, models will make systematic mistakes for some reactions as shown in the manuscript. The framework proposed is a good and clever approach that limits the necessary amount of human intervention. But even with minimal human intervention, will it be feasible for large reaction datasets? Otherwise, it is interesting, but not very practical, and will not receive widespread use. In its current form, I fear that its scope is too narrow. If the authors choose to make LocalMapper more general, maybe it would be possible to include reactions where the ground-truth mappings are known, like QM datasets (there are some for diverse elementary reactions, radical reactions, etc)? This would possibly be a cheap way to add new templates. Or heavily curated datasets with mappings, where a human expert would be able to verify mappings very quickly since most of them are correct. If the authors choose to leave LocalMapper unchanged, they should very clearly describe its scope (organic reactions with templates found in USPTO-50K, and what type of transformations are covered by these templates).

- For non-confident predictions, do the authors recommend to not use them at all (even if that strongly limits the type of reactions LocalMapper can predict)? If they do recommend to use them, there should be some effort to increase the quality of the obtained mappings, to prevent mappings where molecules are ripped apart and undergo dozens of bond-changes (which clearly is wrong).

- On a minor note: The authors should change the code to enable users to optionally run it on the CPU, also (I changed my local version, it's just a few lines) - despite the prediction being slower, many users will want to test the mapper on systems that might not have a GPU readily available.

Reviewer #3 (Remarks to the Author):

The manuscript by Chen et al. explores a graph neural network-based atom-to-atom mapping approach. Atom-mapping is a crucial task to define the reaction center and bond changes in chemical reactions, and atom-mapping information is often required for downstream applications with chemical reactions. It is a very timely and important topic.

The manuscript could be much improved by discussing relevant atom-mapping literature and approaches in the introduction that date back to the 70s. Also, the inaccurate atom maps in the open-source databases motivated certain atom-mapping independent approaches.

I believe that the term uncertainty quantification is misused in the manuscript. The method reports a confidence of 1, whenever the template is in the knowledge base and 0, whenever it is not. It is not a

calibrated uncertainty. Could you rephrase it as a template present in the knowledge base or templates labeled by a human annotator?

The ground-truth atom-maps in the set used come from the NextMove Software NameRXN tool, not Indigo, if the right column of the USPTO-50k dataset was used. NameRXN is of higher quality than Indigo and basically applies templates to the reactions to recover the atom-mapping as a side product. The authors claim that more than 3k are incorrectly labeled in the USPTO-50k dataset, which could be the case. However, I can't find this analysis on the GitHub repository.

What are the manually removed problematic reactions?

How many unique templates/local templates are in the USPTO-50k set?

Are there reactions with multiple matching templates/local templates?

Are there templates/local templates that match multiple functional groups in the same reaction? Or asked differently, how do you ensure that if the same template as the ground truth is returned, the correct atom-mapping was the only possibility to obtain that template? There is a trade-off on how general or specific the templates are. Are there conflicting templates? If such cases exist, how do you prevent the accuracy from being overestimated in your method? Take 12345 as an example. What if it had mapped another oxygen atom in the reactants? Would it have still been counted as correct?

Figure 3e, if I understand correctly, shows a reaction that LocalMapper wrongly mapped but where the template is in their template library (would such examples not decrease the score from the 100% correct? In such cases, the uncertainty calibration would be terrible, as it mistakenly gives the user the impression that it is a very confident (1.0) prediction but is wrong.

There are reactions where leaving groups cannot be determined by the reaction template/local template alone but depend on the reagents. How are they taken into account? It seems that only reactions that are not already in the knowledge base will be sampled in the active learning loop. An example could be the Mitsunobu reaction, where the OH leaving group will be different compared to other reactions (with the same template) but going through a different reaction pathway. That said, the Mitsunobu reactions seemed to be correctly mapped by LocalMapper. However, the same mapping is also given when no DEAD or equivalent is present as a reagent, and the reaction is more ambiguous (e.g., 2082). How do you handle reagents that influence the reaction pathway but are not mapped in themselves?

For 4445/7281, for example, how are you sure that you mapped the correct second reactant and that the atoms do not come from the acetic anhydride instead of the acetic acid?

I would be careful with claims on “perfect accuracy”.

The USPTO-281 is marked as an out-of-distribution reaction set, but comes from the same distribution of patent reactions as the USPTO-50k dataset. How does it perform on the two other datasets that were published by Jaworski et al.? It would be great to include them. Those would be more out-of-distribution than the USPTO-281.

Moreover, there were certain atom-mapping benchmarking studies in the last years on which the final model should be evaluated:

- <https://onlinelibrary.wiley.com/doi/10.1002/minf.202100138>
- <https://pubs.acs.org/doi/abs/10.1021/acs.jcim.2c00344> (Golden dataset)

Please also include results on those datasets.

How fast is LocalMapper? If you map the whole USPTO dataset (more like a real-world example), how does it work there?

The code is open source (under GPL license, MIT or Apache would be preferred as they are more permissive). What seems to be missing are the human annotations that led to the results (the file is empty). It would be great to add that one, and also the trained models and a script to run it. If I wanted to make use of this work, I would like to be able to use it without reannotating reactions or retraining a model.

Could you analyze and comment better on the problematic reactions in USPTO-281 and other datasets in the SI?

The whole part on the analysis of the problematic reactions and comparison to other reaction mappers seems to be missing in the repo. It would be great to add that as well.

A feature that is often missing in atom-mapping tools is the possibility to handle stoichiometry, is this possible with LocalMapper?

In the text, you refer to Fig2e, but it does not exist.

Figure 2, the atom-maps overlap with the molecular structures.

I'm unable to comment on Fig S2, as the quality is too low.

There are plenty of typos, I would suggest using a grammar/English correction tool before resubmitting the manuscript.

Overall, it is an exciting study on a very relevant topic.

REVIEWER COMMENTS

Reviewer #1:

• Doesn't cover benchmarking done before (<https://doi.org/10.1002/minf.202100138>) on a much larger test set. The performance of the algorithm should be reevaluated on a set from the given article.

→ We appreciate the reviewer for this suggestion. We now compare our results on the Golden Dataset in the updated manuscript. However, we found the source of the reactions in this dataset are missing and thus hard to analyze the model performance in terms different reaction sources. Therefore, we also reevaluate LocalMapper on the “train reactions” used in Jarworski et al. (<https://doi.org/10.1038/s41467-019-09440-2>), which is a subset of the Golden dataset but include the source of each reaction including 256 USPTO reactions, 173 typical reactions and 62 mechanistically complex reactions. The results are compared with RXNMapper and GraphormerMapper (<https://doi.org/10.1021/acs.jcim.2c00344>), as shown in Figure 2 in the updated manuscript and attached below.

Table 2. AMM results of RXNMapper, GraphormerMapper, and LocalMapper on manually-mapped reactions collected from four different sources. The highest accuracy and ratio are highlighted in bold font.

Model	Golden	USPTO	Typical	Complex
Accuracy of all predictions				
RXNMapper	86.5%	89.5%	91.3%	59.7%
GraphormerMapper	82.7%	93.8%	87.9%	67.7%
LocalMapper (this work)	89.6%	99.2%	93.6%	69.4%
Accuracy (ratio) of confident predictions				
RXNMapper	95.1% (19.7%)	90.2% (23.8%)	95.7% (13.3%)	50.0% (12.9%)
LocalMapper (this work)	100% (53.3%)	100% (79.7%)	100% (42.8%)	100% (6.5%)

• The article above also introduces an AAM comparison algorithm that does not directly involve the comparison of labeled reaction SMILES strings. While the algorithm's specifics are somewhat challenging to discern from the text, it appears to share similarities.

→ The above paper compared the AAM using condensed graph of reaction (CGR), which we found more robust than comparing reaction template in our previous manuscript. We changed the accuracy comparison using CGRs instead of reaction templates in the updated manuscript.

“The accuracy of AAM predictions is calculated by comparing the condensed graph of reaction (CGR) between the model's prediction and the ground truth using CGRtools toolkit following previous works to ensure that equivalent but different AAMs between the ground truth and model predictions did not lead to underestimations of prediction accuracy.”

• Template-based noise reduction and ELRT look like the method proposed in (<https://doi.org/10.1021/acs.jcim.2c00344>). If yes – what is the difference? If not – the same question.

→ We clarify this question by two points: ELRT (**similar to** the heuristic rules) and noise reduction (**distinct from** AAM fixer proposed in in the mentioned paper, GraphormerMapper).

- ELRT: Although we are not clear how the heuristic rules were made, they are both reaction rules that encode the reaction information into computer readable format.
- Noise reduction: The mechanism of noise reduction used in LocalMapper and GraphormerMapper are different from each other. In LocalMapper, we classify the predicted AAM as unconfident if their extracted ELRT are unknown, whereas in GraphormerMapper the AAM fix the reaction AAM if it detects wrong AAM predictions. In other words, our method focuses on “known correct predictions” and GraphormerMapper focuses on “known incorrect predictions”. Our method actively samples model's blind spot to enhance the model's performance, whereas AAM fixer only fixes the reactions by pre-defined rules and does not improve the model.

To clarify this point, we added the following paragraph in the Discussions Section:

“It is vital to distinguish between LocalMapper's knowledge-based prediction confidence and the AAM fixer employed in GraphormerMapper. While both methods leverage the insights of chemists to enhance prediction accuracy, they diverge significantly in their operational mechanisms. The AAM fixer directly rectifies the model's AAM predictions but doesn't enhance

the model itself. It relies on manual heuristics to correct known inaccuracies, making it challenging to scale up without extensive experimental adjustments. In contrast, knowledge-based prediction confidence empowers human chemists to label uncertain predictions, thereby facilitating model improvement through active learning. This approach is data-driven and easily scalable by expanding active learning iterations. “

• The authors conduct a performance comparison with RXNMapper; however, the article above showcased superior performance compared to RXNMapper. It might be more appropriate to compare against both existing mappers or solely against Graphormer. USPTO267 provided by authors showed 98% accuracy on chytorch-rxnmap installed from PyPI.

→ We thank the reviewer for this suggestion. We now compare the results with both models in the updated manuscript at Table 1 and Table 2.

Table 1. AAM results of RXNMapper, GraphormerMapper, and LocalMapper on USPTO-50K dataset before and after manually checking the reaction AAMs. The best accuracy and ratio are highlighted in bold font. *Accuracy evaluated on the 3,000 reactions sampled from the reactions confidently predicted by both RXNMapper and LocalMapper.

Model	Dataset accuracy		Manual checked accuracy		Calibrated accuracy
	All predictoins	Conf. predictions	Conf. predictions	Conf. ratio	All predictions
RXNMapper	98.1%	99.7%	93.4%*	30.9%	96.1%
GraphormerMapper	92.8%	94.6%*	94.8%*	/	/
LocalMapper (this work)	91.5%	92.8%	100%*	97.0%	98.5%

Table 2. AMM results of RXNMapper, GraphormerMapper, and LocalMapper on manual-mapped reactions examined on four different sources. The highest accuracy and ratio are highlighted in bold font.

Model	Golden	USPTO	Typical	Complex
Accuracy of all predictions				
RXNMapper	86.5%	89.5%	91.3%	59.7%
GraphormerMapper	82.7%	93.8%	87.9%	67.7%
LocalMapper (this work)	89.6%	99.2%	93.6%	69.4%
Accuracy (ratio) of confident predictions				
RXNMapper	95.1% (19.7%)	90.2% (23.8%)	95.7% (13.3%)	50.0% (12.9%)
LocalMapper (this work)	100% (53.3%)	100% (79.7%)	100% (42.8%)	100% (6.5%)

Note that we found that the AAM fixer used in GraphormerMapper was likely optimized for this dataset, so we remove the AAM fixers of GraphormerMapper during evaluation to test the raw model performance. We made this point clear at the first paragraph of “atom-to-atom mapping evaluation”:

“To access the prediction accuracy of LocalMapper, we conducted a comparative analysis with two unsupervised learning-based models: RXNMapper and GraphormerMapper. To ensure a fair model comparison, we evaluate these model without the use of AAM fixer, which automatically fixes the known incorrect AAMs to correct AAMs after predictions. “

• The confidence derived from RXNMapper represents an average attention weight and depends on the number of tokens. Consequently, it necessitates experimentation to ascertain its suitability for intended usage.

1. The relationship between confidence and number of tokens is not specified in the RXNMapper, but similar work using Transformer to predict product (Molecular Transformer) shows the confidence and number of tokens are not necessarily correlated (see Figure 8 of <https://pubs.acs.org/doi/10.1021/acscentsci.9b00576>)
2. The positive correlation between confidence and prediction score of RXNMapper is reported in its SI (see Figure S4 of <https://www.science.org/doi/10.1126/sciadv.abe4166>). Therefore, we consider it is reasonable to compare the confidence score with RXNMapper.

• It’s not clear how the attention block and classifier head are implemented.

→ We now wrote more details of the attention blocks and the classifier in the Method in the updated manuscript. Furthermore, we show the pseudocode of training *LocalMapper* in the Method section as attached below.

Algorithm 1. The pseudocode of training *LocalMapper*.

Algorithm 1: Pseudocode of training *LocalMapper*

Input :

Reactant graph $\mathcal{G}_r(\mathcal{V}_r, \mathcal{E}_r)$; reactant features $\{x_{r,u} \in \mathcal{V}_r, x_{r,uv} \in \mathcal{E}_r\}$;

product graph $\mathcal{G}_p(\mathcal{V}_p, \mathcal{E}_p)$; product features $\{x_{p,u} \in \mathcal{V}_p, x_{p,uv} \in \mathcal{E}_p\}$;

AAM label $y(u_r|u_p)$

Initialize:

Model parameters θ

```
1 for epoch in 1, N do
2    $h_{r,u} \leftarrow MPNN(x_{r,u}, \{x_{r,uv}\})$ 
3    $h_{p,u} \leftarrow MPNN(x_{p,u}, \{x_{p,uv}\})$ 
4    $h_{p,u} \leftarrow MultiHeadAtt(h_{p,u}, \{h_{r,u}\})$ 
5    $\hat{y}(u_r|u_p) \leftarrow Classifier(h_{p,u}, h_{r,u})$ 
6    $loss \leftarrow \mathcal{L}_{AAM}(\hat{y}(u_r|u_p), y(u_r|u_p))$ 
7   update  $\theta$ 
8 end
```

Reviewer #2:

Overall, this is a great piece of work, apart from one major concern: The scope of LocalMapper caused by its use of only USPTO-50K:

- I am concerned about the limited scope of the USPTO-50K dataset underlying the study, as it only comprises a very limited set of chemical reactions, as often shown or highlighted in literature. The authors should demonstrate their work on larger and more diverse datasets, too. Can the mapper be used as-is on larger datasets, such as full USPTO (available free-of-charge), Pistachio, Reaxsys, to name just a few? Or would it require more active learning loops?

→ We thank the reviewer for this critical concern. Using the LocalMapper trained by 1,000 reactions sampled from USPTO-50K, we are able to cover (confidently predict) 544.5K out of 1.06M (51.1%) reactions in the USPTO-full dataset. After sampling additional 1,000 reactions from the full USPTO dataset, we are able to cover 712.6K (66.9%) reactions in the USPTO-full by training a LocalMapper model using 2,000 reactions total (0.2% of total reactions). These results clearly demonstrate that LocalMapper can be used to map large datasets by further sampling reactions from the target datasets. We have updated the discussion section to emphasize this finding:

“Although we labeled only 47.3K reactions (97%) in the USPTO-50K dataset after manually annotating 1,000 reactions in this paper, it’s remarkable that the same model can confidently label 544.5K reactions (51.1%) in the full USPTO dataset. This represents a substantial increase in the labeling efficiency achieved through manual annotation. Furthermore, with two additional active learning iterations, this number grows to 712.6K reactions (66.9%).”

How much manual labelling is estimated to cover those datasets?

→ The lower bound of the manually labeling is the number of unique ELRTs existing in the reaction dataset. Take the USPTO-full dataset (1.06M reactions) mapped by RXNMapper for example, since there are 168,561 unique ELRTs, at least 168,561 manual labeling should be performed to cover all 1.06M reactions in the USPTO-full dataset.

Note that many of the AAMs predicted by RXNMapper may be wrong and yield invalid ELRTs. Out of 168,561 ELRTs extracted from 1.06M RXNMapper-mapped reactions, 133,381 ELRTs only appear once, which are highly likely to be wrong. If we consider ELRTs appearing more than 5 times, 11,339 manual labeling is required to map 81.9% of the reactions in USPTO-FULL dataset.

Will LocalMapper be actively developed and ready to use on larger datasets in the near future?

→ Yes, as we showed the LocalMapper trained using additional 1,000 reactions sampled from the USPTO-full dataset in this revision. This version of LocalMapper is able to confidently map 712.6K (66.9%) reactions. The model and mapped dataset by this version of LocalMapper is available on GitHub, and we will train the model for more iterations (hopefully achieve 90% coverage) after the method is fixed upon the paper acceptance.

- The accuracy of the confident predictions of LocalMapper are indeed very good. When I tested the code on different datasets (following the authors suggestion to compare templates instead of raw mappings), the confident predictions were right in all cases, and revealed a few errors in eg. USPTO-1k-TPL. Thus, the 100% accuracy seems to hold indeed, which is great!

However, the ratio of confident predictions gets lower, the more the reactions differ from what is found in USPTO-50k. For example, on a subset of USPTO-1k-TPL <https://doi.org/10.1038/s42256-020-00284-w> (450K USPTO reactions with templates that occur more than 100 times), I find 9% not confident but correct mappings, and 4% not confident and incorrect mappings. So, although still USPTO, the confidence ratio is now only 87%. For those that are not confident and incorrect, LocalMapper provides VERY (like ripping apart molecules) wrong mappings in some cases, so it seems to not generalize well to reactions beyond the known templates in some cases.

For quantum-mechanical datasets of reactions (where the ground-truth mapping is known), I find for a subset of the wb97xD3 dataset by Grambow et al <https://doi.org/10.1038/s41597-020-0460-4> (12K elementary reactions), 58% not confident but correct templates, and 38% not confident and incorrect mappings. Thus, the confidence ratio is now only 4%. Sure, the accuracy of confident mappings is 100%, but if 96% of my dataset results in unconfident predictions, this is not very helpful. Further moving away from USPTO-50k, when I test on enzymatic reactions, namely MetaCyc (where MetaCyc mappings do have some flaws, but most of them are correct), I find 40% not confident but correct, and 57% not confident and incorrect predictions, with a confidence ratio of 3%. This highlights the very limited scope of USPTO-50k that the authors rely on, with a sharp drop in the confidence ratio, as well as the ratio of correct templates for unconfident predictions. I think the authors need to put more work in either making LocalMapper more general, or changing the manuscript to highlight that the given confidence ratio is only indicative for the narrow set of USPTO-50K, and people need to retrain it on their dataset (making its use less attractive possibly).

→ We thank the reviewer very much for the thorough examination. The generalizability of LocalMapper to a completely different dataset is indeed the major limitation of this approach. As the reactions from different domains can vary a lot, it is not surprising to see low confidence ratio on different datasets (4% on quantum-mechanical reactions and 3% on enzymatic reactions) since they may follow completely different reaction mechanisms (all unimolecular one-to-many reactions for quantum-mechanical reactions and enzyme-catalyzed reactions for enzymatic reactions). In these cases, we suggest users to label and retrain the model by their domain knowledge, since the quality of the model is highly sensitive to the labeling chemists.

It is important to note that this limitation (low confidence ratio on other domains) is at the same time an advantage of giving a warning of potential low prediction accuracy, where the model may require more active-learning iterations before being used for AAM prediction. This is an important information especially for tasks requiring high precision such as AAM. We now discuss this important point in the Discussion Section of the updated manuscript.

“However, it’s important to note that LocalMapper tends to yield a significantly lower ratio of confident predictions when applied to entirely distinct reaction datasets, such as quantum-mechanical reactions and enzymatic reactions. These reactions may follow entirely different reaction mechanisms (unimolecular one-to-many reactions and enzyme-catalyzed reactions, respectively) that were never encountered during active learning on organic reaction datasets. For such cases, we recommend engaging domain-specific chemists to undertake active learning iterations to adapt the model effectively before using it for large-scale AAM tasks.”

Also, we modified the paper title from "Precise Atom-to-Atom Mapping via Human-Machine Collaboration" to "Precise Atom-to-Atom Mapping for Organic Reactions via Human-Machine Collaboration" to clarify the scope of this paper.

- The above issue is incredibly hard to tackle: With a human in the loop, the covered reactions will necessarily be limited. Without human intervention, models will make systematic mistakes for some reactions as shown in the manuscript. The framework proposed is a good and clever approach that limits the necessary amount of human intervention. But even with minimal human intervention, will it be feasible for large reaction datasets?

→ The coverage of a given dataset is highly dependent on the domain of the reaction dataset since each new ELRT discovered in the dataset needs to be manually labeled. If the reactions in the target dataset are similar to the training dataset, LocalMapper can confidently map over 50% of the reactions on a dataset containing over 1 million reactions (USPTO-full).

Otherwise, it is interesting, but not very practical, and will not receive widespread use. In its current form, I fear that its scope is too narrow. If the authors choose to make LocalMapper more general, maybe it would be possible to include reactions where the ground-truth mappings are known, like QM datasets (there are some for diverse elementary reactions, radical reactions, etc)? This would possibly be a cheap way to add new templates. Or heavily curated datasets with mappings, where a human expert would be able to verify mappings very quickly since most of them are correct. If the authors choose to leave LocalMapper unchanged, they should very clearly describe its scope (organic reactions with templates found in USPTO-50K, and what type of transformations are covered by these templates).

→ We thank the reviewer for this suggestion. In the current form, we want to focus the scope of LocalMapper on organic reactions. Lin et al. (<https://doi.org/10.1002/minf.202100138>) provided a set of accurately atom-mapped reactions (including 1,851 reactions), which is a good reaction source to make LocalMapper more generalized to other type of reactions.

- For non-confident predictions, do the authors recommend to not use them at all (even if that strongly limits the type of reactions LocalMapper can predict)?

→ If the user is aiming 100% correct AAM, such as making a template-based retrosynthesis model, we would suggest the users not to use the non-confident predictions at all. Even if the non-confident predictions may have 50% chance to be correct, the other 50% will be noisy data and making the downstream model learn the wrong chemistry.

If they do recommend to use them, there should be some effort to increase the quality of the obtained mappings, to prevent mappings where molecules are ripped apart and undergo dozens of bond-changes (which clearly is wrong).

→ The disconnected atom-mapping can be solved by increasing the neighbor weight of the atom-mapping function (https://github.com/shuan4638/LocalMapper/blob/main/scripts/atom_mapper.py). However, there is a trade-off between focusing on the neighbor weight and original predictions. If the neighbor weight is set too high, the algorithm would act like maximum common subgraph and always map the neighboring atoms in the reactants, causing obviously incorrect AAM such as mapping the wrong oxygen for hydrolysis reactions shown in the Figure 3.

- On a minor note: The authors should change the code to enable users to optionally run it on the CPU, also (I changed my local version, it's just a few lines) - despite the prediction being slower, many users will want to test the mapper on systems that might not have a GPU readily available.

→ We thank the reviewer for this suggestion. We now updated the code on GitHub to enable CPU usage.

s

Reviewer #3:

The manuscript could be much improved by discussing relevant atom-mapping literature and approaches in the introduction that date back to the 70s. Also, the inaccurate atom maps in the open-source databases motivated certain atom-mapping independent approaches.

→ We thank the reviewer for the insightful comment. We now elaborated the introduction section to include more literatures as follows:

“Existing methods for AAM identification can be generally categorized into rule-based and ML-based methods. Most of the rule-based methods identify AAM based on minimal chemical distance (MCD) or maximum common subgraph (MCS) isomorphism algorithms. Solving the AAM problem by rule-based approaches is challenging because such a subgraph isomorphism problem has been known to be an NP-hard problem since the 1970s, and there is no efficient algorithm to find the exact solutions. Recently, machine learning models have been developed to bypass the time-consuming subgraph matching process and map the atoms in the reactions directly by the information extracted from the model’s learned features. Schwaller et al. proposed an unsupervised-learning-based model called RXNMapper to link the grammar dependency of each atom between reactants and products. By focusing on specific attention weights of the language model, RXNMapper not only achieved a promising prediction accuracy that outperformed existing rule-based methods but also largely reduced the computational time of performing AAM on large reaction datasets. More recently, Nugmanov et al. developed GraphomerMapper using a similar unsupervised learning strategy with RXNMapper based on a graph-based Transformer and trained the model on a much larger reaction dataset.”

, and added an introducing figure in the introduction section to emphasize the motivation of this work.

Figure 1. The importance of accessing correct atom-to-atom-mapping (AAM) in terms of generating retrosynthesis reaction templates and deriving reaction mechanisms. The upper AAM is generated by the proposed AAM model *LocalMapper*, and the bottom AAM is recorded in the USPTO reaction dataset.

I believe that the term uncertainty quantification is misused in the manuscript. The method reports a confidence of 1, whenever the template is in the knowledge base and 0, whenever it is not. It is not a calibrated uncertainty. Could you rephrase it as a template present in the knowledge base or templates labeled by a human annotator?

→ We thank the reviewer for the suggestion. We now changed the term “uncertainty quantification” to “knowledge-based prediction confidence” to avoid the confusion between our method and conventional approach of uncertainty quantification.

The ground-truth atom-maps in the set used come from the NextMove Software NameRXN tool, not Indigo, if the right column of the USPTO-50k dataset was used.

→ The atom-maps are generated by Indigo software as described in the original paper of USPTO-50K (<https://doi.org/10.1021/acs.jcim.6b00564>, first paragraph in Data Sets):

“First, all reactions were atom mapped using the Indigo toolkit. (22) The input reactions to the Indigo mapper originated directly from text mining which provides the preassignment of some reagents (mainly solvents or catalysts which are unambiguous). To improve these mappings further the Indigo mapper was tweaked to ignore both charges and valency and to allow bond order changes.”

, and introduced in RXNMapper (<https://doi.org/10.1126/sciadv.abe4166>, third paragraph in Introduction):

“Most public reaction data come with rule-based Indigo atom-maps (17), which are taken as ground truth for subsequent work (18–23), irrespective of the explicit warnings about atom-maps quality issues (24).”

NameRXN is of higher quality than Indigo and basically applies templates to the reactions to recover the atom-mapping as a side product. The authors claim that more than 3k are incorrectly labeled in the USPTO-50k dataset, which could be the case. However, I can't find this analysis on the GitHub repository.

→ We now add the incorrectly mapped reactions in GitHub repository. In particular, it could be found at https://github.com/kaist-amsg/LocalMapper/tree/main/data/USPTO_50K/view_problematics.ipynb. Also, we added six examples in Fig. S2 for reference.

Reaction type	Invalid product map	Confusing reagent
Example 1		
Example 2		
Example 3		

Figure S2. Three examples of reactions with (left) invalid product and (right) confusing reagent in the USPTO-50K dataset. The molecules causing the problems are highlighted in red.

What are the manually removed problematic reactions?

1. *Invalid product map*: products having duplicated or missing atom-map
2. *Confusing reagent*: existence of product-like reagents (Tanimoto similarity ≥ 0.5)
3. *Missing reactant*: reactions with missing reactants (hydrolysis reactions)

→ We note that 3. *missing reactant* is hard to be addressed before extracting the ELRT, so we now only exclude the reactions belonging to the first two categories to simplify the process. The examples of the two types of problematic reactions are added in the Supporting Information and shown in the previous response.

How many unique templates/local templates are in the USPTO-50k set?

→ Since there are several wrong AAMs in the original USPTO-50k dataset (mapped by Indigo), it is hard to tell how many unique templates there are if correct AAMs are provided. If we simply use the dataset AAMs as ground truth, there are 1,102 unique templates in USPTO-50k dataset.

Are there reactions with multiple matching templates/local templates?

→ Yes. Although most of atom-mapped reactions will only yield one matching reactions template, ring forming reactions sometimes give different reaction templates indicating the same reaction due to the failure of canonicalizing the reaction SMARTS.

Are there templates/local templates that match multiple functional groups in the same reaction? Or asked differently, how do you ensure that if the same template as the ground truth is returned, the correct atom-mapping was the only possibility to obtain that template? There is a trade-off on how general or specific the templates are. Are there conflicting templates? If such cases exist, how do you prevent the accuracy from being overestimated in your method? Take 12345 as an example. What if it had mapped another oxygen atom in the reactants? Would it have still been counted as correct?

→ The reviewer is correct that there is a trade-off on how general or specific the templates are, and it is necessary to find the sweet spot when designing the template extraction algorithm. The reaction template should be specific enough for chemists to understand what reaction it is, at the same time the template should be general enough to map the same template to different reactions sharing the same reaction mechanism. This is exactly the reason why we expand the local reaction template (LRT, <https://pubs.acs.org/doi/10.1021/jacsau.1c00246>, only include changed atoms) with extended chemical informative groups.

The ELRT of reaction 12345 is C-C-[O:1]-C=O>>[O:1]-C=O, indicating a demethylation reaction on the ester oxygen, which is chemically not likely to occur. If the oxygen is mapped on the water (the correct mapping), the ELRT will be [O:1].C-C-O-[C:2]=O>>O=[C:2]-[O:1], indicating a chemically reasonable ester hydrolysis reaction.

To prevent the method overestimating the method, we changed the evaluation method to comparing the condensed graph of reactions (CGRs) as used in previous AAM works (<https://doi.org/10.1002/minf.202100138> and <https://doi.org/10.1021/acs.jcim.2c00344>) instead of comparing reaction templates.

Figure 3e, if I understand correctly, shows a reaction that LocalMapper wrongly mapped but where the template is in their template library (would such examples not decrease the score from the 100% correct? In such cases, the uncertainty calibration would be terrible, as it mistakenly gives the user the impression that it is a very confident (1.0) prediction but is wrong.

→ The example in Figure 3e shows the LocalMapper wrongly mapped the reaction, and it does *not* give the correct template recorded in the template library. All reactions shown in Figure 3 are uncertain predictions, where the templates extracted from predicted atom-mappings are not in template library. Therefore, the 100% accuracy for confident predictions holds.

We notice this result can be confusing, therefore we replace this result by the percentage of statistics of correctly predicted confident predictions instead.

There are reactions where leaving groups cannot be determined by the reaction template/local template alone but depend on the reagents. How are they taken into account? It seems that only reactions that are not already in the knowledge base will be sampled in the active learning loop. An example could be the Mitsunobu reaction, where the OH leaving group will be different compared to other reactions (with the same template) but going through a different reaction pathway. That said, the Mitsunobu reactions seemed to be correctly mapped by LocalMapper. However, the same mapping is also given when no DEAD or equivalent is present as a reagent, and the reaction is more ambiguous (e.g., 2082). How do you handle reagents that influence the reaction pathway but are not mapped in themselves?

→ This is a good point. Because many of the reactions in the USPTO dataset do not contain necessary reagent information (only 87.2% of the reactions in USPTO-50k dataset do not include any reagent, and many of them do not include necessary reagents), we assume the necessary reagents exist if the reaction is recognizable. E.g. the ELRT of reaction 2082 is O-[C:1].[O:2]-c>>[C:1]-[O:2], which can be achieved by existence of DEAD/DIAD and triphenylphosphine.

We are aware that only 170 (39.4%) out of 431 of the reactions sharing this ELRT in the original USPTO-50k dataset include both DEAD and equivalent reagents. Similarly, 1,166 (49.4%) out of 2,362 Suzuki coupling reactions do not contain Pd, the important catalyst for the reaction work. If we consider the reagent information in the ELRT and only consider the prediction as confident, the rest of the reactions (60.6%) cannot be confidently labeled despite the clear information given in the ELRT. Therefore, we simply assume all necessary reagents exist in the work. We now make this point explicit in the Method Section:

“Due to the absence of essential reagent and catalyst information in many reactions within the USPTO-50K dataset, we do not incorporate reagent and catalyst details into the reaction templates. For instance, we observed that at least 1,166 (49.4%) out of 2,362 Suzuki coupling reactions lack Pd catalyst, 520 (34.6%) out of 1,500 nitro reduction reactions do not feature a reduction agent, and 170 (39.4%) out of 431 Mitsunobu reactions do not include diethyl azodicarboxylate (DEAD) or diisopropyl

azodicarboxylate (DIAD). As a result, we make the simplifying assumption that common and necessary reagents or catalysts are present in the reactions during template extraction.”

For 4445/7281, for example, how are you sure that you mapped the correct second reactant and that the atoms do not come from the acetic anhydride instead of the acetic acid?

→ If the atoms are mapped from acetic anhydride, the ELRT is [N:2].C-C(=O)-O-[C:1]=O>>O=[C:1]-[N:2]. If the atoms are mapped from acetic acid, the ELRT is [N:2].O-[C:1]=O>>O=[C:1]-[N:2]. These two different AAMs will give different reaction template and thus easy to be distinguished.

I would be careful with claims on “perfect accuracy”.

→ We fully agree that claiming the perfect accuracy without really looking into all the predictions (49,996 reactions) can be misleading. Therefore, we state in the updated manuscript that this accuracy is evaluated using randomly sampled 3,000 reactions from confident predictions generated by RXNMapper and LocalMapper:

“To investigate the incorrectly predicted confident predictions from RXNMapper and LocalMapper, we randomly sampled 3,000 reactions from a pool of 14,422 reactions confidently predicted by both RXNMapper and LocalMapper. After manually checking these predicted AAMs, we found all the confident predictions from LocalMapper are indeed correct, but they have been incorrectly mapped in the original dataset.”

The USPTO-281 is marked as an out-of-distribution reaction set, but comes from the same distribution of patent reactions as the USPTO-50k dataset. How does it perform on the two other datasets that were published by Jaworski et al.? It would be great to include them. Those would be more out-of-distribution than the USPTO-281.

Moreover, there were certain atom-mapping benchmarking studies in the last years on which the final model should be evaluated:

- <https://onlinelibrary.wiley.com/doi/10.1002/minf.202100138>
- <https://pubs.acs.org/doi/abs/10.1021/acs.jcim.2c00344> (Golden dataset)

Please also include results on those datasets.

→ We appreciate the reviewer for this suggestion. We now compare our results on the Golden Dataset in the updated manuscript. However, we found the source of the reactions in this dataset are missing and thus hard to analyze the model performance in terms different reaction sources. Therefore, we also reevaluate LocalMapper on the “train reactions” used in Jarworski et al. (<https://doi.org/10.1038/s41467-019-09440-2>), which is a subset of the Golden dataset but include the source of each reaction including 256 USPTO reactions, 173 typical reactions and 62 mechanistically complex reactions. The results are compared with RXNMapper and GraphormerMapper (<https://doi.org/10.1021/acs.jcim.2c00344>), as shown in Figure 2 in the updated manuscript and attached below.

Table 2. AMM results of RXNMapper, GraphormerMapper, and LocalMapper on manually-mapped reactions collected from four different sources. The highest accuracy and ratio are highlighted in bold font.

Model	Golden	USPTO	Typical	Complex
Accuracy of all predictions				
RXNMapper	86.5%	89.5%	91.3%	59.7%
GraphormerMapper	82.7%	93.8%	87.9%	67.7%
LocalMapper (this work)	89.6%	99.2%	93.6%	69.4%
Accuracy (ratio) of confident predictions				
RXNMapper	95.1% (19.7%)	90.2% (23.8%)	95.7% (13.3%)	50.0% (12.9%)
LocalMapper (this work)	100% (53.3%)	100% (79.7%)	100% (42.8%)	100% (6.5%)

How fast is LocalMapper? If you map the whole USPTO dataset (more like a real-world example), how does it work there?

In our experiment, it takes 35 minutes to map the USPTO-50K dataset (49,996 reactions), and around 14 hours for the full USPTO dataset (1.06M reactions) by LocalMapper. We now add this information in the Method Section:

“Training LocalMapper takes 4 hours for the USPTO-50K dataset (at 5th iteration) and 6 hours for the full USPTO dataset (at 2nd iteration), while the inference takes 35 minutes for the former and 14 hours for the the latter reaction datasets. “

The code is open source (under GPL license, MIT or Apache would be preferred as they are more permissive). What seems to be missing are the human annotations that led to the results (the file is empty). It would be great to add that one, and also the trained models and a script to run it. If I wanted to make use of this work, I would like to be able to use it without reannotating reactions or retraining a model.

→ All the files (including annotation results, trained models and scripts) were uploaded before the manuscript submission, and successfully tested by reviewer #2.

Could you analyze and comment better on the problematic reactions in USPTO-281 and other datasets in the SI?

→ As we now evaluated the models on the subset of Golden dataset, these reactions are fixed by the author of Golden dataset (<https://doi.org/10.1038/s41467-019-09440-2>). Thus, we removed the description of problematic reactions in USPTO-281 in SI.

The whole part on the analysis of the problematic reactions and comparison to other reaction mappers seems to be missing in the repo. It would be great to add that as well.

→ We now upload the analysis of problematic reactions at https://github.com/kaist-ams/LocalMapper/tree/main/data/USPTO_50K/view_problematics.ipyn, and share the comparison to other reaction mappers in GitHub repository.

A feature that is often missing in atom-mapping tools is the possibility to handle stoichiometry, is this possible with LocalMapper?

→ We believe that handling incorrect stoichiometry in the reaction is beyond the scope of AAM problem, while it requires adding necessary reactant information that does not exist in the reaction. While this can be simply done by adding missing elements in the reactant side as reported in Jarworski et al. (<https://doi.org/10.1038/s41467-019-09440-2>), this approach does not seem to be chemically feasible. Take the reaction below for example, if the source of atoms colored in red in the product is missing in the reactant side, the method proposed by Jarworski et al. simply adds a methane and a water in the reactant side, which is unlikely to work for this reaction. A better candidate of the additional reactant could be phosgene (carbonyl dichloride) or diphosgene (trichloromethyl chloroformate), however, handling this issue may be too far from the AAM problem.

Instead of handling incorrect stoichiometry, LocalMapper can “detect” incorrect stoichiometry since the predicted AAMs from LocalMapper would not give chemist-verified reaction template if the stoichiometry in the given reaction is wrong, and the reactions will be classified as unconfident prediction.

Figure:

In the text, you refer to Fig2e, but it does not exist.

→ We thank the reviewer for pointing this out. We now fix this issue and make sure there is no similar mistake in the updated manuscript.

Figure 2, the atom-maps overlap with the molecular structures.

→ We thank the reviewer for pointing this out. We now fix the atom-maps in Figure 2

I’m unable to comment on Fig S2, as the quality is too low.

→ As we now compare the AAMs using condensed graph of reaction (CGR), Figure S2 in the SI has been removed.

There are plenty of typos, I would suggest using a grammar/English correction tool before resubmitting the manuscript.

→ We thank the reviewer for pointing this out. We fixed these typos and had a thorough check for the updated the manuscript.

REVIEWER COMMENTS

Reviewer #1 (Remarks to the Author):

The revised manuscript covers the literature well and shows a valuable contribution to atom-to-atom mapping task solving by proper benchmarking. The authors added a simple python interface for mapping reactions, which increased the value for the community of the work made.

Reviewer #2 (Remarks to the Author):

The authors have addressed some of my concerns, and raised a few more. I believe this is a good piece of work, but it must be demonstrated a bit better (still).

Point 1a: The coverage of USPTO-full (51% without further active learning, and 67% with a few thousand more labels) is indeed encouraging. Minor point: The number of reactions in the "two additional active learning iterations" should be added to the manuscript (it only appears in the author response currently).

Point 1b+c: The amount of manual labeling required for larger datasets is very high, and the authors indicate that de-facto EVERY template needs to be hand-labeled once, and that the generalization ability of LocalMapper is therefore low. This makes me again question the practicability of the approach: If LocalMapper does not generalize well to new templates, it would be important to show that LocalMapper is at least much faster than e.g. exhaustive application of the list of 1000 templates. How long would it take to apply the templates from the hand-labeled reactions to USPTO-50k (which yields the correct atom mappings) vs the 35min reported for LocalMapper? Are there cases where more than one template generates the correctly mapped reactions? I expect this analysis to be much in favor of LocalMapper, but think it would be important to show.

Point 2+3: Although I would have liked the authors to expand the scope of LocalMapper, they at least now highlight the applicability domain of their approach, which I think to be sufficient.

Point 4: I think the authors may have missed an opportunity here to increase the quality of LocalMapper for non-confident predictions. They could have e.g. added a check how many bonds are broken or formed for a suggested mapping, and for clearly wrong mappings then e.g. automatically increased the neighbor weight or defaulted to some other algorithm. This is also in line with Point 1b+c above: if the

novelty of LocalMapper only relies on the confident predictions recovering a known template, why not just apply templates in the first place? That being said, for the "Complex dataset", e.g. the confident ratio is only 6%, so it would not be practical to only use the confident predictions/simply apply templates (and the unconfident predictions perform surprisingly well, so there actually is a good ability of LocalMapper to generalize under these circumstances). I therefore imagine it difficult to judge when an unconfident prediction would be ok to use and when not. However, the additional benchmarks look encouraging, so maybe the "very wrong" mappings I encountered before were only for datasets where LocalMapper is not applicable. Did the authors look into the wrong mappings for the additional benchmarks, and checked e.g. bond breaking/forming statistics?

Point 5: Thanks for adding CPU support.

Reviewer #3 (Remarks to the Author):

The manuscript and code base have much improved since the last version. It is excellent work and deserves to be published.

A few remaining minor comments:

- The abstract should also contain the word "confident" when writing about the "3,000 randomly sampled reactions".
- Atom-mapping is helpful but not enough to derive the reaction mechanism (Fig1 and abstract).
- The GitHub repo states that the raw data for USPTO 50k was downloaded from rxnmapper. The raw_data.csv file, which is referred to in the Jupyter notebook, was removed. The data provenance there is not clear to me.
- The authors write about a cleaned dataset with 48,830 reactions for AAM evaluation. Where is this dataset?
- In the checked dataset of 3000 reactions, the ester hydrolysis and similar reactions are wrong in RXNMapper (great catch!). But there seems to be at least one mistake. For instance, for 12189, you report False, False, True, but the RXNMapper mapping seems to be equivalent to the LocalMapper mapping.

12189,Br[C:4]([CH:2]([CH3:1])[Br:3])=[O:5].CCN(CC)CC.Cc1cccc1.[OH:6][CH2:7][CH2:8][O:9][CH2:10][c:11]1[cH:12][cH:13][cH:14][cH:15][cH:16]1>>[CH3:1][CH:2]([Br:3])[C:4](=[O:5])[O:6][CH2:7][CH2:8][O:9][CH2:10][c:11]1[cH:12][cH:13][cH:14][cH:15][cH:16]1,[CH2:8]([CH3:7])[N:9]([CH2:10][CH3:11])[CH2:12][CH3:13].[O:5]=[C:4]([CH:2]([Br:3])[CH3:1])[Br:6].[cH:16]1[cH:17][cH:18][cH:19][cH:20][c:15]1[CH3:14].[cH:28]1[cH:27][c:26]([cH:31][cH:30][cH:29]1)[CH2:25][O:24][CH2:23][CH2:22][OH:21]>>[cH:28]1[cH:27][c:26]([cH:31][cH:30][cH:29]1)[CH2:25][O:24][CH2:23][CH2:22][O:5][C:4](=[O:21])[CH:2]([Br:3])[CH3:1],Br[C:1]([CH:2]([CH3:3])[Br:4])=[O:5].CCN(CC)CC.Cc1cccc1.[OH:6][CH2:7][CH2:8][O:9][CH2:10][c:11]1[cH:12][c

H:13][cH:14][cH:15][cH:16]1>>[C:1]([CH:2]([CH3:3])[Br:4])(=[O:5])[O:6][CH2:7][CH2:8][O:9][CH2:10][c:11]1[cH:12][cH:13][cH:14][cH:15][cH:16]1,False,False,True.

Is this an error by the human expert?

- It looks like back when the USPTO 50k dataset was published, the ester hydrolysis and similar reactions were also wrongly mapped by NameRXN. I want to reiterate that if the data was taken from RXNMapper, the original mapping was not from Indigo (as claimed in the response) but NameRXN (<https://pubs.acs.org/doi/full/10.1021/acs.jcim.6b00564>, in the SI, DataSetB, “rxnSmiles_Mapping_NameRxn” column).
- How was the “prediction confidence score of 0.9” as a threshold for RXNMapper chosen? Is this the best-performing threshold according to the introduced metrics?
- It would be great to specify that you are comparing an unsupervised approach (e.g., RXNMapper) against a supervised approach (LocalMapper). Is it that surprising to surpass an unsupervised approach when between 40 and 80% of the reaction templates of the test set were part of the training?

Reviewer #2:

Point 1a: The coverage of USPTO-full (51% without further active learning, and 67% with a few thousand more labels) is indeed encouraging. Minor point: The number of reactions in the "two additional active learning iterations" should be added to the manuscript (it only appears in the author response currently).

→ The corresponding sentence was added in the main text as well, for which we kindly ask the reviewer refer to the following sentence in our last revision (before Table 2):

"To enhance LocalMapper's ability to confidently predict a wider spectrum of organic reactions, we conducted further training of the model for 2 additional iterations on the USPTO-FULL dataset (containing 1,065,119 reactions) with sampling 500 reactions at each iteration (k=500, n=2). All the reactions in this test set were excluded from the training set of LocalMapper."

Point 1b+c: The amount of manual labeling required for larger datasets is very high, and the authors indicate that de-facto EVERY template needs to be hand-labeled once, and that the generalization ability of LocalMapper is therefore low. This makes me again question the practicability of the approach: If LocalMapper does not generalize well to new templates, it would be important to show that LocalMapper is at least much faster than e.g. exhaustive application of the list of 1000 templates.

→ Before discussing the generalizability of LocalMapper, it is important to recognize the two different generalizability involved in this context: model generalizability and template generalizability.

1. Model generalizability: how well LocalMapper can correctly predict the AAMs of the reactions that do not exist in the training set (e.g. accuracy of an unseen test set).
2. Template generalizability: how many reactions can one reaction template represent (e.g. the ratio of confident prediction for unseen dataset).

As none of the reactions recorded in the typical reactions and complex reactions exist in the train set of LocalMapper, the higher accuracy compared to RXNMapper and GraphormerMapper shown in Table 2 does not imply a "low generalizability" of LocalMapper.

We assume the reviewer is more likely referring to the template generalizability, where nearly 2,000 reaction templates can only cover 67% of the USPTO-full dataset and 6.5% of the complex reactions. While it is possible to generalize the reaction template by removing functional groups (or even atom symbols) in the reaction templates like the templates used in LocalRetro (ref.14) and LocalTransform (ref. 32), ignoring these details would significantly downgrade the resolution of the given chemical reaction and potentially yield wrong AAMs. Take the esterification reaction shown in Figure S1a (shown below) for example, the local reaction template (LRT, reaction template without functional groups) of this reaction could also be understood as an SN2 reaction or Mitsunobu reaction. As the oxygen #2 of this esterification reaction should come from the alcohol group (left molecule), this mapping can only be confident when the carbonyl group information is encoded in the reaction template. In other words, although using LRT can hugely generalize these three reaction types (esterification, SN2, and Mitsunobu) into one reaction template, the information of LRT is not sufficient for confident AAM. Thus, reaction templates, in our opinion, should be defined differently depending on the nature and purpose of the tasks.

How long would it take to apply the templates from the hand-labeled reactions to USPTO-50k (which yields the correct atom mappings) vs the 35min reported for LocalMapper?

→ If the reviewer is referring to generating AAM for a given reaction by two steps (without ML models):

- (1) apply all the known reaction templates to the reactant
- (2) keep the products matching the original product

, applying all the known templates extracted from the hand-labeled reactions to all reactions in USPTO-50k takes around 6 hours to finish, which is 10 times longer than the LocalMapper prediction. We have added the description of the potential cost of applying all reaction templates to USPTO-50k in the second paragraph of the Methods:

"...For comparison, applying all known reaction templates directly to the reactants to obtain the reaction AAM by matching the known product takes approximately 10 times longer, requiring 6 hours to complete on the USPTO-50k dataset, in contrast to the 35 minutes required by LocalMapper."

Are there cases where more than one template generates the correctly mapped reactions? I expect this analysis to be much in favor of LocalMapper, but think it would be important to show.

→ Yes, there are cases where more than one template can generate the same correctly mapped reactions. However, it does not mean one correctly mapped reaction will generate multiple reaction templates. Take the amide formation below as example:

There are 8 reaction templates that lead the reactant react to the correct product, but there is only one of the reaction templates (the one in red box) will be extracted from the original reaction due to the specific leaving group in the reaction.

Point 4: I think the authors may have missed an opportunity here to increase the quality of LocalMapper for non-confident predictions. They could have e.g. added a check how many bonds are broken or formed for a suggested mapping, and for clearly wrong mappings then e.g. automatically increased the neighbor weight or defaulted to some other algorithm.

→ We thank the reviewer for the suggestion. In fact, we have tried this idea during the last revision, but it turned out that this approach would decrease the model accuracy, especially when predicting AAMs for complex reactions. It is ambiguous to define how many broken/formed bonds can be defined as “clearly wrong mappings”. For example, the chart below shows the number of bonds broken/formed for the reactions in each subset in Golden dataset according to their ground truth AAM. If we simply define AAMs yielding more than 10 bonds broken/formed, prediction results on 27.4% of the complex reactions may change from correct to incorrect. However, we are grateful to the reviewer for this valuable suggestion.

This is also in line with Point 1b+c above: if the novelty of LocalMapper only relies on the confident predictions recovering a known template,

→ Generating confident predictions recovering the known template is NOT the only novelty of LocalMapper, but one of them:

- (1) LocalMapper is the first supervised-learning based ML model for AAM prediction.
- (2) LocalMapper is the first ML model that can generate and guarantee extremely high accurate reaction AAM.
- (3) The active learning framework allows LocalMapper to generate unseen reaction templates and later learned from them after verified or fixed by the human chemist.

Making confident predictions recovering a known template is only one of the main contributions of this paper (which facilitates novelty #2).

why not just apply templates in the first place?

→ We thank the reviewer for the kind suggestion, but there are two obstacles if applying templates in the first place:

1. As the templates are simplified by removing hydrogen and charge information, many reactions cannot reach their original product by simply applying reaction templates. In our experiment, we found there are 10,151 reactions that cannot reach their original product by applying reaction templates directly to their reactants.
2. As shown in previous comments, applying all the known templates to obtain reaction AAM can be very time-consuming (taking x10 times than LocalMapper prediction).

Nonetheless, we appreciate the reviewer for the valuable comment, and we will consider adopting this idea in our future work.

Did the authors look into the wrong mappings for the additional benchmarks, and checked e.g. bond breaking/forming statistics?

→ We thank for the reviewer's suggestion. We did look into the cases where LocalMapper gave wrong mappings for the additional benchmarks, but we did not analyze them according to the bond/forming statistics. We agree with the reviewer that analyzing these results can present more comprehensive statistics for the additional benchmarks. Therefore, we added the results of analyzing the bond breaking/forming statistics at Figure 5 and added a paragraph describing these results:

Figure 5: Comparative analysis of the number of bond changes in reaction AAMs. The figure illustrates the number of bond changes derived from the AAMs of (a) patent, (b) typical, and (c) complex reactions, incorrectly predicted by LocalMapper, RXNMapper, and GraphormerMapper (GraphMapper) compared with the ground truth (GT) AAMs. Additionally, examples of complex reactions with AAM predictions from LocalMapper are presented, showcasing instances where the predicted AAM yields (d) more, (e) the same, or (f) fewer number of bond changes compared to the ground truth AAMs. Ground truth AAMs and corresponding bond change count for the example reactions are annotated in black, while those generated by LocalMapper are highlighted in red.

“In Figure 5a, 5b, and 5c, we conduct a detailed analysis of the number of bond changes derived from the AAMs (according to their CGRs) of reactions that were inaccurately predicted by LocalMapper, RXNMapper, and GraphormerMapper, respectively. Generally, the majority of incorrectly predicted AAMs across all three models and various reaction sources result in an increased number of bond changes compared to the corresponding ground truth AAMs. Notably, RXNMapper stands out for producing a substantial number of incorrectly predicted AAMs that result in a decreased number of bond changes in patent reactions, primarily involving ester hydrolysis reactions. To illustrate the impact of such predictions, we present an example in Figure 5d, wherein even a small number (4) of incorrectly predicted AAMs can result in a significantly higher count (10) of bond changes compared to the correct AAMs. However, the examples depicted in Figure 5e and 5d underscore that predicted AAMs, showing either the same or fewer bond changes, do not consistently align with the ground truth AAMs, especially in the context of complex reactions.”

Reviewer #3:

The manuscript and code base have much improved since the last version. It is excellent work and deserves to be published.

A few remaining minor comments:

- The abstract should also contain the word “confident” when writing about the “3,000 randomly sampled reactions”.

→ The full sentence of the referring part in abstract is

“More importantly, the confident predictions given by LocalMapper, which cover 97% of 50K reactions, show 100% accuracy for 3,000 randomly sampled reactions.”

We believe this statement is sufficiently clear to specify the prediction accuracy for confident predictions rather than all predictions.

- Atom-mapping is helpful but not enough to derive the reaction mechanism (Fig1 and abstract).

→ We agree with the reviewer that atom-mapping alone is helpful but not enough to derive reaction mechanism without prior chemistry knowledge. Thus, we now modified the corresponding sentence to “AAM..., which is important for deriving the reaction mechanism based on chemical knowledge” in the abstract, and “...and deriving reaction mechanisms based on chemical knowledge” in Fig 1.

- The GitHub repo states that the raw data for USPTO 50k was downloaded from rxnmapper. The raw_data.csv file, which is referred to in the Jupyter notebook, was removed. The data provenance there is not clear to me.

→ We thank the reviewer for the comment. We now uploaded raw_data.csv file to the corresponding repository.

- The authors write about a cleaned dataset with 48,830 reactions for AAM evaluation. Where is this dataset?

→ For the fully reproducible results, we keep the raw_data.csv file including 50k reactions in the GitHub repo and show how the 1,166 problematic reactions are filtered out in view problematics.ipynb. The remaining 48,830 reactions can be obtained after running the code. Because we regard this as a post-processing work only for model evaluation, we did not release this dataset.

- In the checked dataset of 3000 reactions, the ester hydrolysis and similar reactions are wrong in RXNMapper (great catch!). But there seems to be at least one mistake. For instance, for 12189, you report False, False, True, but the RXNMapper mapping seems to be equivalent to the LocalMapper mapping.

12189,Br[C:4]([CH:2]([CH3:1])[Br:3])=[O:5].CCN(CC)CC.Cc1cccc1.[OH:6][CH2:7][CH2:8][O:9][CH2:10][c:11]1[cH:12][cH:13][cH:14][cH:15][cH:16]1>>[CH3:1][CH:2]([Br:3])[C:4](=[O:5])[O:6][CH2:7][CH2:8][O:9][CH2:10][c:11]1[cH:12][cH:13][cH:14][cH:15][cH:16]1.[CH2:8]([CH3:7])[N:9]([CH2:10][CH3:11])[CH2:12][CH3:13].[O:5]=[C:4]([CH:2]([Br:3])[CH3:1])[Br:6].[cH:16]1[cH:17][cH:18][cH:19][cH:20][c:15]1[CH3:14].[cH:28]1[cH:27][c:26]([cH:31][cH:30][cH:29]1)[CH2:25][O:24][CH2:23][CH2:22][OH:21]>>[cH:28]1[cH:27][c:26]([cH:31][cH:30][cH:29]1)[CH2:25][O:24][CH2:23][CH2:22][O:5][C:4](=[O:2]1)[CH:2]([Br:3])[CH3:1],Br[C:1]([CH:2]([CH3:3])[Br:4])=[O:5].CCN(CC)CC.Cc1cccc1.[OH:6][CH2:7][CH2:8][O:9][CH2:10][c:11]1[cH:12][cH:13][cH:14][cH:15][cH:16]1>>[C:1]([CH:2]([CH3:3])[Br:4])=[O:5][O:6][CH2:7][CH2:8][O:9][CH2:10][c:11]1[cH:12][cH:13][cH:14][cH:15][cH:16]1,False,False,True.

Is this an error by the human expert?

→ We thank the reviewer for the careful check. Yes, this is an error caused when validating by us. We checked all the predictions carefully again and did not find any other mislabeled reaction. We have updated the comparison file and reflected the updated accuracy at Table 1 in the revised manuscript (Conf. accuracy from 93.4% to 93.6%, calibrated accuracy from 96.1% to 96.2%).

- It looks like back when the USPTO 50k dataset was published, the ester hydrolysis and similar reactions were also wrongly mapped by NameRXN. I want to reiterate that if the data was taken from RXNMapper, the original mapping was not from Indigo (as claimed in the response) but NameRXN (<https://pubs.acs.org/doi/full/10.1021/acs.jcim.6b00564>, in the SI, DataSetB, “rxnSmiles_Mapping_NameRxn” column).

→ We thank the reviewer for the valuable information. After reading the original paper again we realized that the original papers used both Indigo and NameRXN for atom-mapping, while the dataset used in this paper, taken from RXNMapper, was originally mapped from NameRXN, not Indigo. Although we do not have a specific figure or sentence mentioning the source of the original mapping of USPTO-50k dataset, we deeply appreciate the reviewer’s efforts in correcting this important fact.

- How was the “prediction confidence score of 0.9” as a threshold for RXNMapper chosen? Is this the best-performing threshold according to the introduced metrics?

→ The prediction score of 0.9 was selected according to the best accuracy of confidence score of 0.9 shown in Figure S4 of the original paper (<https://doi.org/10.1126/sciadv.abe4166>). To clarify this point, we added the following statement in the manuscript when introducing the prediction confidence score of 0.9:

“Because RXNMapper also gives a confidence score for each prediction, which shows a positive correlation with the prediction accuracy¹⁸, we binarize the confident score of RXNMapper by its prediction confidence score of 0.9 (according to the best

performing results shown in the Supplementary Material of¹⁸) to facilitate the comparison with the confident predictions generated by LocalMapper.”

- It would be great to specify that you are comparing an unsupervised approach (e.g., RXNMapper) against a supervised approach (LocalMapper). Is it that surprising to surpass an unsupervised approach when between 40 and 80% of the reaction templates of the test set were part of the training?

→ In the first paragraph of the discussions section, we did mention the major difference between LocalMapper and existing models lies in the use of chemist-labeled data during training.

It is not surprising to see LocalMapper predicts better than unsupervised learning approaches when 40 to 80% of the reaction templates of the test reactions were part of the training. Nonetheless, the fact that the model trained on 2K high-quality (manually labeled) reactions works better than the model trained on 1M (RXNMapper) and over 10M unlabeled reactions (GraphormerMapper) emphasize the importance of manual labeling for model training.

REVIEWERS' COMMENTS

Reviewer #2 (Remarks to the Author):

The authors have addressed all my concerns. Thank you for checking the timing (LocalMapper vs simple template application), template applicabilities and bond change statistics for wrong mappings - these results are all in favor of LocalMapper.